# Effect of Renewable Fuels and Intake O₂ Concentration on Diesel Engine Emission Characteristics and Reactive Oxygen Species (ROS) Formation

**Louise Gren** [1,2] , **Vilhelm B. Malmborg** [1,2] , **Nicklas R. Jacobsen** [3] , **Pravesh C. Shukla** [4,†],
**Katja M. Bendtsen** [3] , **Axel C. Eriksson** [1,2] , **Yona J. Essig** [5] , **Annette M. Krais** [5] ,
**Katrin Loeschner** [6] , **Sam Shamun** [4] , **Bo Strandberg** [5] , **Martin Tunér** [4] , **Ulla Vogel** [3] and
**Joakim Pagels** [1,2,*]

1   Ergonomics and Aerosol Technology, Lund University, SE-22100 Lund, Sweden;
    louise.gren@design.lth.se (L.G.); vilhelm.berg_malmborg@design.lth.se (V.B.M.);
    axel.eriksson@design.lth.se (A.C.E.)
2   NanoLund, Lund University, 22100 Lund, Sweden
3   National Research Centre for the Working Environment, DK-2100 Copenhagen, Denmark;
    nrj@nfa.dk (N.R.J.); katjabendtsen@gmail.com (K.M.B.); UBV@nfa.dk (U.V.)
4   Division of Combustion Engines, Lund University, SE-221 00 Lund, Sweden; pravesh@iitbhilai.ac.in (P.C.S.);
    samshamun@gmail.com (S.S.); martin.tuner@lth.lu.se (M.T.)
5   Division of Occupational and Environmental Medicine, Lund University, SE-22100 Lund, Sweden;
    julie.essig@med.lu.se (Y.J.E.); annette.krais@med.lu.se (A.M.K.); bo.strandberg@med.lu.se (B.S.)
6   National Food Institute, Technical University of Denmark, DK-2800 Kgs. Lyngby, Denmark;
    kals@food.dtu.dk
*   Correspondence: joakim.pagels@design.lth.se
†   Current affiliation: Department of Mechanical Engineering, Indian Institute of Technology Bhilai,
    Chhattisgarh 492015, India.

**Abstract:** Renewable diesel fuels have the potential to reduce net $CO_2$ emissions, and simultaneously decrease particulate matter (PM) emissions. This study characterized engine-out PM emissions and PM-induced reactive oxygen species (ROS) formation potential. Emissions from a modern heavy-duty diesel engine without external aftertreatment devices, and fueled with petroleum diesel, hydrotreated vegetable oil (HVO) or rapeseed methyl ester (RME) biodiesel were studied. Exhaust gas recirculation (EGR) allowed us to probe the effect of air intake $O_2$ concentration, and thereby combustion temperature, on emissions and ROS formation potential. An increasing level of EGR (decreasing $O_2$ concentration) resulted in a general increase of equivalent black carbon (eBC) emissions and decrease of $NO_x$ emissions. At a medium level of EGR (13% intake $O_2$), eBC emissions were reduced for HVO and RME by 30 and 54% respectively compared to petroleum diesel. In general, substantially lower emissions of polycyclic aromatic hydrocarbons (PAHs), including nitro and oxy-PAHs, were observed for RME compared to both HVO and diesel. At low-temperature combustion (LTC, $O_2 < 10\%$), CO and hydrocarbon gas emissions increased and an increased fraction of refractory organic carbon and PAHs were found in the particle phase. These altered soot properties have implications for the design of aftertreatment systems and diesel PM measurements with optical techniques. The ROS formation potential per mass of particles increased with increasing engine $O_2$ concentration intake. We hypothesize that this is because soot surface properties evolve with the combustion temperature and become more active as the soot matures into refractory BC, and secondly as the soot surface becomes altered by surface oxidation. At 13% intake $O_2$, the ROS-producing ability was high and of similar magnitude per mass for all fuels. When normalizing by energy output, the lowered emissions for the renewable fuels led to a reduced ROS formation potential.

**Keywords:** EGR; RME; HVO; ROS; soot; PAHs; aerosol

## 1. Introduction

Renewable diesel fuels can replace petroleum-based diesel in compression ignition (CI) engines (diesel engines) in order to reduce net $CO_2$ and particulate matter (PM) emissions to the atmosphere [1–4]. Diesel exhaust gases such as $NO_x$ and volatile organic species (VOCs) are precursors to smog and secondary aerosol formation [5]. The PM composition determines the radiative forcing (RF), with Black Carbon (BC) contributing to positive radiative forcing by absorbing incoming solar radiation. BC from on-road diesel exhausts has been estimated to contribute to around 10% of the atmospheric BC burden, with a RF of 44 mW/m$^2$ [6]. The exposure to diesel exhaust emissions (PM and gases such as $NO_x$) has been related to adverse health impacts such as various lung and cardiovascular diseases [7–9] where the PM fraction has specifically been attributed to lung cancer in animal models [10].

Oxidative stress is one established biological response related to the toxic potential of PM [11] which is initiated when high levels of reactive oxygen species (ROS) overwhelm the antioxidant defence systems within cells. High levels of ROS in cells can trigger a cascade of events associated with inflammation and apoptosis [12] but also induce oxidative DNA damage, mutations, and pathways towards carcinogenesis. Several studies have shown that a high specific surface area (SSA) of insoluble particles is a key property for toxicological responses, and in particular for the generation of ROS [13,14].

Diesel exhaust PM consists primarily of solid, insoluble carbonaceous particles with high SSA (closely related to BC), an organic fraction including polycyclic aromatic hydrocarbons (PAHs) and an ash fraction enriched with metals [15]. All of these components have been identified to contribute to toxicologically relevant responses in general and to ROS formation specifically [16–18]. This has motivated further studies, and in particular, a need to identify what particle properties drive diesel PM toxicity.

Renewable diesel emissions, such as those from hydrotreated vegetable oil (HVO) and rapeseed methyl ester (RME), have shown both higher [19–21] and lower [19,22] ROS formation potential compared to petroleum diesel emissions. Modifications to the engine combustion conditions can induce large differences in the composition and characteristics of emitted PM. Control of the engine operating conditions is therefore essential for a just comparison of fuel effects on PM properties. It remains unclear if the contradictory results of earlier studies are a result of engine type and operation or a fundamental difference that can be linked to the fuel replacement.

RME is a fatty acid methyl ester (FAME) fuel produced by transesterification of rapeseed oil. It has an oxygen content of about 10%. HVO consists of paraffinic hydrocarbons, chemically similar to petroleum diesel but with a negligible aromatic content and a higher cetane number [23,24]. Previous studies show a 50–80% PM mass reduction with FAME fuels [1,2] and 20–50% reduction for HVO [3,4]. When PM mass emissions were reduced, increased nucleation mode particle number emissions have been observed [25]. Emission characteristics have also been found to depend on the feedstock from which the fuel was derived [26–28]. This includes examples of altered primary particle size and changes to the internal nanostructure depending on FAME substitution level and oxygen content of the FAME fuels [29,30].

Exhaust gas recirculation (EGR) is a well-known and commonly used technique that reduces combustion temperatures in order to reduce $NO_x$ emissions from engines [31]. The effects of EGR and fuel have been widely studied on regulated gas emissions such as $NO_x$, CO and hydrocarbons (HC), as well as on engine performance and soot emissions. $NO_x$ emissions decrease with increasing EGR (decreasing $O_2$ concentration) while HC, CO and soot emissions increase [32,33]. However, to our knowledge, only a few studies exist on the effects that EGR has on particle composition and characteristics [34–38]. The chemical composition, such as PAH and metal emissions, and the physical

properties of particle emissions induced by varying the EGR and thus combustion conditions is of particular interest for understanding the drivers of diesel PM toxicity.

Low temperature combustion (LTC) summarizes combustion concepts where flame temperatures are drastically reduced to simultaneously achieve very low $NO_x$ and PM emissions from engines. It is known from basic combustion studies and model flames that soot properties change with decreasing combustion temperature. Non-regulated emissions with health impact such as aromatic compounds [39], including PAHs, may increase as combustion temperature decreases [40].

We utilized an experimental heavy-duty diesel engine with precise control of the combustion parameters to investigate how the ROS formation potential and properties of PM depend on the level of EGR and on the fuel formulation. The aim of this work was to quantify the influence of combustion temperature on emission characteristics and particle-induced ROS formation for petroleum diesel and renewable HVO and RME fuels. Based on the results, we present a novel framework of the relation between combustion conditions, particle properties, and the ROS formation potential. In addition, we describe a method to generate and collect engine exhaust particles designed to have varied physicochemical characteristics for future in vivo studies.

## 2. Materials and Methods

### 2.1. Emission Generation

The experiments were performed with a modern 6-cylinder 13-L compression ignition engine, modified to operate with a single cylinder. The engine was equipped with a compressed air line, a backpressure valve and an exhaust gas recirculation (EGR) system to re-circulate a well-defined proportion of the exhaust gas back to the cylinder. Detailed engine specifications can be found in [41]. The engine was operated with three fuels at a fixed low engine load of $IMEP_g$ 6 bar (gross indicated mean effective pressure) and an engine speed of 1200 RPM in all experiments. The fuels tested were Swedish ultra-low sulphur diesel (referred to as petroleum diesel) without addition of biofuels (MK1; B0) and two renewable diesel fuels: hydrotreated vegetable oil (HVO) and rapeseed methyl ester (RME). All fuels were tested without blending.

To allow a direct comparison of emissions between the fuels, the combustion phasing (CA50 position) was held constant at five crank angle degrees (CAD) after top dead center (ATDC), which was achieved by fine tuning the start of injection (SOI) timing. This CA50 position is near the optimum condition where maximum efficiency and reduced emissions can be achieved for the tested engine. The engine was equipped with an XPI common rail injection system with an injector with 10 holes and 148° spray angle. The common rail pressure was set at 1200 bar for all the test conditions throughout the experiments.

In a first stage, EGR sweeps were carried out for each fuel to probe the impact of EGR level (and thereby combustion temperature) on the emission characteristics. In these tests, the averaging time was ~1.5 min and the EGR level was increased in steps (10–12 steps for each fuel), starting with no EGR (ambient $O_2$ concentration of 21%) until the $O_2$ concentration in the engine intake was reduced to ~9%. With no EGR, the global Lambda (λ, excess air ratio) was ~4.6 for all tested fuels. λ was reduced to ~1.2 when the $O_2$ concentration in the engine intake was reduced to 9%.

Six operation points were chosen for repeated longer experiments, with averaging time >30 min (Table 1). A more extensive analysis of five of these points was carried out allowing quantitative comparisons of the emissions between the fuels. These were for petroleum diesel: ~10% intake $O_2$ (LTC), ~13% intake $O_2$ ($NO_x$ reducing EGR) and ~17% intake $O_2$ (low EGR). All three fuels, i.e., diesel, RME and HVO were sampled at ~13% intake $O_2$.

**Table 1.** A summary of the longer replicate experiments, including number of replicates, mean exhaust gas recirculation (EGR) level (intake $O_2$ concentration), mean equivalence ratio and fuel specifications. The number of replicates is the number of the filter collections of particulate matter (PM), the number of filters pooled and extracted for offline chemical analyses and reactive oxygen species (ROS) formation potential are denoted within brackets. No offline analyses were performed of diesel samples collected at 10.8% intake $O_2$.

| | Diesel | | | | HVO | RME |
|---|---|---|---|---|---|---|
| Average intake $O_2$ concentration (% ± 1 std. dev.) | 9.7 ± 0.1 | 10.8 ± 0.5 | 13.1 ± 0.2 | 16.7 ± 0.4 | 13.3 ± 0.2 | 12.8 ± 0.5 |
| Equivalence ratio ± 1 std. dev. | 0.80 ± 0.00 | 0.73 ± 0.03 | 0.60 ± 0.00 | 0.43 ± 0.02 | 0.59 ± 0.01 | 0.65 ± 0.03 |
| Number of replicates | 3 (3) | 3 (0) | 6 (2) | 3 (3) | 2 (2) | 5 (2) |
| Calorific value (MJ/kg) | 43.15 | | | | 44.1 * | 37.3 ** |
| Stoichiometric ratio | ~14.49 | | | | ~14.90 | ~12.3 ** |

\* Neste renewable diesel handbook, Neste Corporation. \*\* RME values from [42].

## 2.2. Dilution System

The raw exhaust emissions were sampled from the exhaust pipe before any aftertreatment device with a heated 6 mm stainless steel probe (150 °C), followed by dilution in two stages to roughly simulate atmospherically relevant dilution and relevant partitioning of PAHs. The first dilution stage was performed with two ejector dilutors in parallel diluted with heated air (150 °C) to 1:4. The exhaust was further diluted with filtered room temperature air to a total dilution factor of 1:30–150 (dependent on the experiment) in a stainless steel partial flow dilution tunnel (Figure A1 in Appendix A).

## 2.3. Emission Characterization

The measurement setup is described in Figure A1 (Appendix A). The raw gas emissions of $NO_x$, HC, $O_2$, CO and $CO_2$ were measured in samples from the exhaust line with an AMA i60 (AVL, Graz, Austria) emission system. The equivalent Black Carbon (eBC, "soot") mass concentration in the undiluted exhaust was measured with a photoacoustic technique (Micro Soot Sensor model 483, AVL, Graz, Austria), and used for quantification of eBC. This measure is often used as an online indicative measure of particulate matter (PM) mass emissions from diesel engines [43]. With the abovementioned raw emission data, fuel flow and in-cylinder pressure measurement the gross indicated specific emissions were calculated by the means of chemical equilibrium, and power output calculations. The full procedure of obtaining the indicated emissions is explained in [44].

The diluted exhaust emissions were characterized after the dilution tunnel using a soot particle aerosol mass spectrometer (SP-AMS, Aerodyne Inc. Billerica, MA, USA), a fast mobility particle analyzer (model DMS500, Cambustion Ltd., Cambridge, UK) and an aethalometer (model AE33, Magee Scientific, Ljubljana, Slovenia). The absorption wavelength of 880 nm was used to estimate the eBC concentrations measured by the aethalometer, using the standard settings of particle optical properties recommended by the manufacturer. The $CO_2$ in the diluted exhaust was measured with a non-dispersive infrared $CO_2$ analyser (LI-8020, LI-COR, Lincoln, Dearborn, MI, USA). The dilution ratio was monitored with the $CO_2$ measurements in the raw and diluted exhaust. In order to reduce the concentrations to those in the optimal range for the SP-AMS and aethalometer, a third dilution stage where a part of the exhaust gas was first led through a HEPA-filter to remove particulates was used specifically for these two instruments.

### 2.3.1. Thermal-Optical Carbon Analysis and Transmission Electron Microscopy Imaging

Samples for thermal optical analysis of organic carbon (OC) and elemental carbon (EC) were collected on quartz filters (Pallflex Tissuequartz, 47 mm) and analysed with a thermal optical analyzer (Model 2001, DRI, Chicago, IL, USA) using the EUSAAR_2 protocol [45]. In addition, extracted samples (described in Section 2.4) were analysed for the relative fractions of OC and EC. The OC fraction was divided into non-refractory OC (nrOC) and refractory OC (rOC). The nrOC consists of carbon

that evolved at temperatures up to 450 °C in inert gas (OC1 to OC3 in the EUSAAR_2 protocol) and refractory OC of carbon that evolved at 650 °C in inert gas and at 500 °C in a 2% $O_2$ gas mixture (OC4 and Pyrolytic Carbon in the EUSAAR_2 protocol).

To analyse the soot particle aggregate structure and primary particle size, samples were collected with electrostatic precipitation using a nanometer aerosol sampler (model 3089, TSI Inc., Shoreview, MN, USA) on lacey carbon coated Cu-grids and analysed with a transmission electron microscopy (JEOL 3000F). The TEM was operated at 300 kV and equipped with a Schottky FEG and $2 \times 2$ k CCD.

The TEM images were analysed with the software ImageJ for primary particle size determination. The diameter of clear primary particles without overlap at the edges of the soot agglomerates was measured in TEM images with magnification of minimum 20,000. For two samples, Printex90 (P90) and diesel particles collected at 10% intake $O_2$ (Diesel 10%), partly overlapping primary particles had to be included in order to acquire a large enough number of counted particles. The number of analysed particles varied between 43–167 primary particles, the number of agglomerates varied between 2–17 for the various samples. The sample size of the number of measured primary particles corresponds to a 95% confidence interval. A lognormal distribution was fitted to the primary particle size distribution and the geometric mean was used as the mean primary particle size.

The specific surface area (SSA; $m^2$/g) for each measured primary particle was estimated by using the diameter ($d_{pp}$) and an assumed inherent material density ($\rho_{pp}$) of 1.8 g/$cm^3$ according to the following equation [46]:

$$SSA = \frac{6}{\rho_{pp} \cdot d_{pp}}$$

By assuming point contact between the primary particles in the agglomerates, we estimate that the primary particle SSA is representative as the sample SSA. The primary particle SSA distribution was assumed to be lognormal and the geometric mean was used as an estimate of the sample SSA.

### 2.3.2. SP-AMS

The SP-AMS is an online instrument that determines that particle chemical composition by sampling the aerosol directly from diluted exhaust. It was operated with a time-resolution of 10 s. The SP-AMS was used to investigate and quantify the chemical composition of the non-refractory organic and refractory carbonaceous PM. To investigate the non-refractory composition, the SP-AMS was run in the standard aerosol mass spectrometer (AMS) mode with vaporization using only thermal desorption on a heated tungsten target (600 °C), followed by electron ionization (70 eV) and detection in a high-resolution time-of-flight mass spectrometer. The particle phase total PAH concentration was calculated from parent ions between 202 Th and 300 Th [47]. For investigation of refractory components, we used the "dual-vaporizer" configuration [48], in which a 1064 nm intracavity laser is added for vaporization of rBC. We calibrated the instrument using 300 nm (mobility equivalent) diameter ammonium nitrate (standard AMS mode) and carbon black (Regal Black, Cabot Corp., Alpharetta, GA, USA) particles (dual vaporizer mode). A detailed description of the detection and analysis methods can be found in Appendix A.

### 2.4. PM Collection with HVCI and Extraction for ROS, PAH and Metal Analysis

Samples from the five chosen operation points were collected with a High-Volume Cascade Impactor (HVCI 900, BGI by Mesa Labs, Butler, NJ, USA) on Teflon filters followed by methanol extraction. There were three operating points for petroleum diesel (EGR levels corresponding to 10%, 13%, 17% intake $O_2$) and one operating point each for RME and HVO (13% intake $O_2$). In order to collect sufficient mass of particulates on the sample filter for analysis and future toxicological studies, each engine operating point could be run for as long as 3 h. PM samples from the dilution tunnel were collected on the final filter of the HVCI with a flow rate of 900 lpm. The cut-off size was 1 μm (PM1) and the final filter (Whatman PTFE, 150 mm, pore size 5 μm) was loaded with 10–20 mg. At least two particle filters were pooled in each case during the extraction. The filters were left to equilibrate

in room temperature (RT) over night before weighing the collected masses. The filters were stored in −80 °C before extraction. After sample collection, the PTFE filters were extracted by sonication (3 × 30 min) in analytical grade methanol (<25 °C), followed by low-pressure evaporation (150 mbar, <35 °C) in 10 mL glass vials [49]. The glass vials were left overnight to equilibrate in RT before weighing, both before and after evaporation. The gravimetric mass extraction efficiency was for all samples ≥85%.

### 2.4.1. ROS Assay

The ability of the extracted PM to generate ROS was determined using the cells-free version of the 2′,7′-dichlorodihydrofluorescein diacetate (DCFH$_2$-DA) assay as previously described in greater detail [50,51]. However, in the current protocol, all PM was tested for autofluorescence where the ROS probe was replaced by Hank's balanced saline solution (HBSS). Printex90, a commercially available carbon black, was used as a reference material and was analysed along with the extracted PM samples. Briefly, the DCFH$_2$-DA (#D399, Invitrogen, Waltham, MA, USA) was chemically hydrolysed in the dark with NaOH to generate 2′,7′-dichlorodihydrofluorescein (DCFH$_2$), which was further diluted with phosphate buffer (pH 7.4) to 0.04 mM. The PM suspensions were prepared using 16 min sonication (Branson S-450D) in Hank's balanced saline solution (HBSS, without phenol, #H6648, Sigma Aldrich, St. Louis, MO, USA). The samples were further diluted in HBSS and tested at 0 μg/mL and eight doubling PM concentrations from 1.05 up to 101.25 μg/mL. The final concentration of DCFH$_2$ at assay start was 0.01 mM. Generated ROS caused formation of 2′,7′-dichlorofluorescein (DCF) from DCFH$_2$ that was spectrofluorimetrically measured following 3h of incubation in the dark (37 °C and 5% CO$_2$). Excitation and emission wavelengths were $\lambda_{ex}$ = 490 nm and $\lambda_{em}$ = 520 nm, respectively (Victor Wallac-2 1420; PerkinElmer, Skovlunde, Denmark).

### 2.4.2. PAH Analysis

The PM samples in the glass vials and quality controls were extracted and cleaned in accordance with a previously described procedure [52] with some modifications. Full description of the PAH extraction and analytical purification procedures can be found in Appendix A. The concentration of 32 native PAHs (including the 16 US EPA priority PAHs) and 16 alkylated species, 17 nitrated and 9 oxygenated PAHs (nitro-PAHs and oxy-PAHs) and 6 dibenzothiophenes (DBTs) were analysed. Full description of compounds investigated, the transitions, limits of detection (LODs), as well as retention times (RT) for all targets are shown in Tables A1–A3 in Appendix A. In short, prior to extraction two labelled internal standard mixtures containing 16 deuterated U.S. EPA priority PAHs, and four deuterated nitro-PAHs were spiked to the samples. Particles were extracted with 3 mL dichloromethane for 3 h, using a Sonica Ultrasonic Extractor (Soltec, Milan, Italy). The samples were cleaned using silica columns prior to concentration to a final volume of approx. 30–40 μL under nitrogen flow. Target compounds were separated on an Agilent 5975C mass spectrometer (MS) coupled to a 7890A gas chromatograph (GC, Agilent Technologies, Santa Clara, CA, USA). Electron impact ionization (EI) was performed for PAHs, alkylated PAHs and DBTs. Electron capture negative chemical ionization mode (ECNCI) was used for the nitro- and oxy-PAH species. The MS was operated in selected ion monitoring mode (SIM) for both EI and ECNCI modes.

### 2.4.3. Metal Analysis

The extracted PM samples were analysed for metal content by ICP-MS as previously described [53] but with slightly modified extraction times. It was not possible to transfer the amount of ≤1 mg particle matter from the glass vials used in the filter extraction procedure to containers suitable for microwave-assisted acid digestion to completely degrade the PMs. Therefore, acid extractions were performed by adding a volume of 1 mL of 25% (v/v) nitric acid directly to the vials. Samples were first agitated at 600 oscillations per min overnight (Stuart Scientific SF1 shaker), then incubated for approximately 7 h at room temperature without agitation and then shaken for another 72 h and

transferred with 6 mL of ultrapure water into polypropylene tubes. Before analysis, the samples were centrifuged for 5 min at 4500 × g (Heraeus Multifuge X3 FR, Thermo Scientific, Waltham, MA, USA), because incomplete digestion of the particles was achieved. A volume of 5 mL of the supernatant was transferred to a new polypropylene tube and diluted 5-fold with 5% nitric acid. A triple quadrupole inductive coupled plasma mass spectrometer (ICP-MS) (Agilent 8900 ICP-QQQ, Santa Clara, CA, USA) equipped with a MicroMist borosilicate glass concentric nebulizer and a Scott type double-pass water-cooled spray chamber was run in no gas (Cd, Hg, Pb, Bi, U) or helium (V, Cr, Mn, Fe, Co, Ni, Cu, Ga, As, Se, Rb, Sr, Ag, In, Cs, Ba, Tl) mode with 0.1–3 s integration time per mass. Quantification was performed based on external calibration. The results should be considered semi-quantitative.

## 3. Results and Discussion

### *3.1. Exhaust Emissions*

#### 3.1.1. Equivalent Black Carbon and Regulated Emissions

Figure 1 displays the exhaust emission factors (mass/kWh) of equivalent black carbon (eBC), $NO_x$ and gas-phase hydrocarbons (HC) for the three fuels as a function of exhaust gas recirculation (EGR). At combustion without EGR (i.e., at 21% intake $O_2$), the $NO_x$ emissions were the highest for all fuels while the eBC emissions were low, less than 4 mg/kWh (Figure 1a). The eBC emissions were substantially reduced for HVO and RME compared to petroleum diesel at $O_2 < 14\%$. The $NO_x$ emissions decreased with the decreasing intake $O_2$ concentration while eBC increased. This BC–$NO_x$ trade-off has previously been attributed to the decrease in flame temperature induced by a lowered $O_2$ concentration. The decreased temperature efficiently reduces the formation of $NO_x$, but also decreases the soot oxidation (removal) rate which results in increased soot emissions [54,55]. The BC–$NO_x$ trade-off was observed for all fuels, in agreement with the literature [31,32]. Very high levels of EGR (low intake $O_2$ concentration) resulted in low temperature combustion (LTC), with simultaneously low eBC and $NO_x$ emissions. The combustion efficiency is still high at LTC, meaning that almost all fuel is combusted. Having high EGR values (corresponding to 9–10% $O_2$ concentration) may result in almost no $NO_x$ emissions, due to the reduction of peak in-cylinder temperatures below the threshold level of 1800 K for $NO_x$ formation. However, although a low $O_2$ concentration and combustion temperature resulted in low emissions of both PM and $NO_x$, the combustion became more incomplete and CO and HC emissions increased considerably (Figure A2a in Appendix A and Figure 1b).

The fuel effect on $NO_x$ emissions was small, and our data of the $NO_x$ emissions of RME do not confirm some previous studies where fatty acid methyl ester (FAME) fuels were reported to yield a small increase (up to 20%) in $NO_x$ emissions (reviewed in [2]). In addition, the fuel effects on the eBC emissions are less clear in the case without EGR (at 21% intake $O_2$), which is in agreement with a previous study on HVO and petroleum diesel using the same engine and fuel without EGR [41]. The HC emissions were similar for petroleum diesel and HVO but reduced for RME at low ($O_2 > 15\%$) and very high levels of EGR ($O_2 < 11\%$).

The effect of fuel properties was studied in more detail at 13% intake $O_2$ with repeated and longer time-series (>1 h). This operation point was chosen for its relevance to emissions high in eBC (and elemental carbon, EC) and a comparably low influence of lubricating oil derived PM. Repeated measurements at 13% intake $O_2$ showed that eBC was reduced by 30%, (95% CI (14%, 46%)) and 54% (95% CI (44%, 69%)), for HVO and RME, respectively.

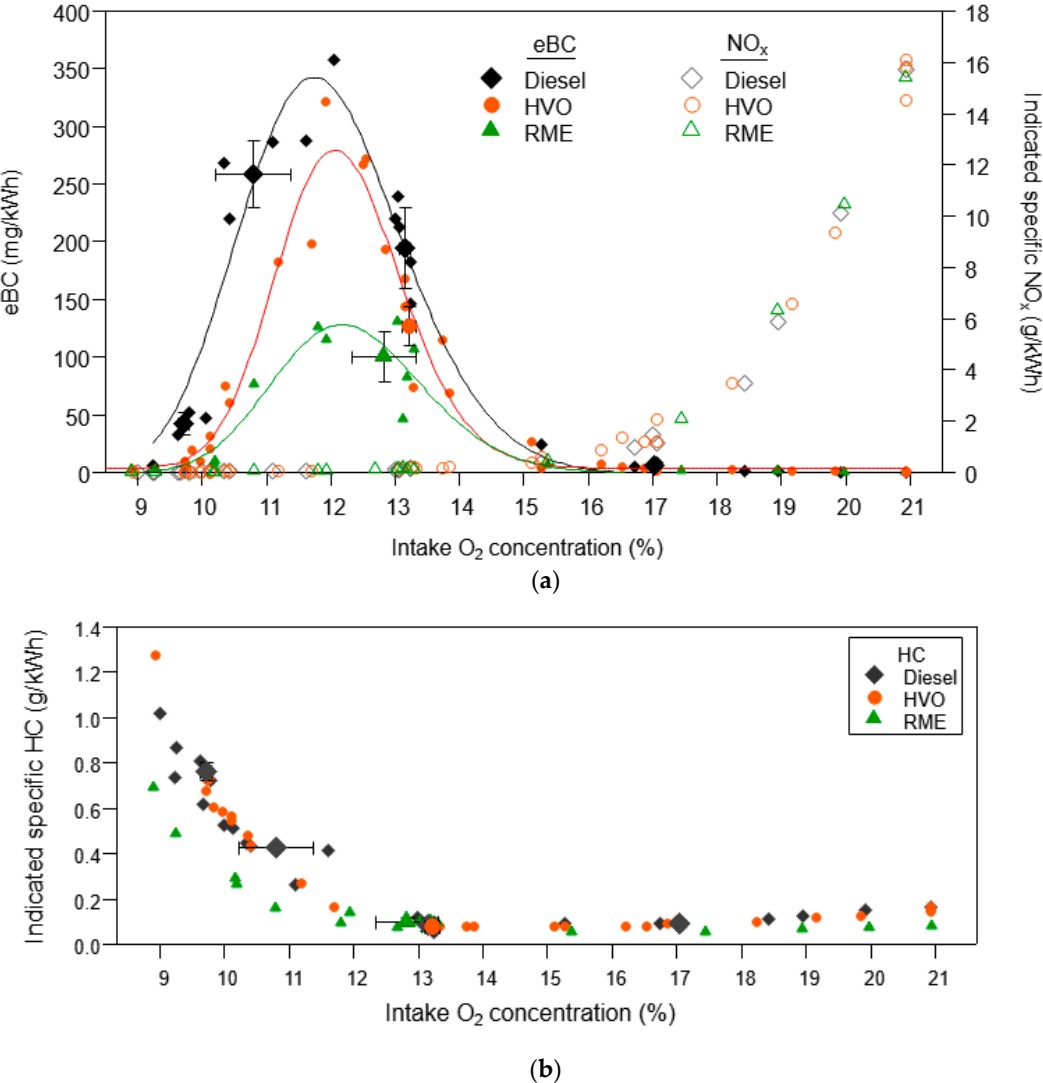

**Figure 1.** Emission factors (mass/kWh) for equivalent black carbon (eBC), $NO_x$ and HC for all fuels, shown as a function of exhaust gas recirculation (EGR). EGR is expressed as intake $O_2$ concentration. The lines shown in (**a**) are lognormal fits to all eBC data points of respective fuel. Combustion temperature increases with increasing intake $O_2$ concentrations. The eBC and $NO_x$ emission factors as a function of EGR are displayed in (**a**). The emission factors of HC are shown in (**b**) and were well correlated with the emission factors of CO (Figure A2a). The HC–CO correlation is high for all fuels (Figure A2b, R > 0.9). The data in Figure 1 are from EGR sweeps with relatively short averaging time in each point (~1.5 min) and from longer filter collections (replicates). Markers with error bars (±1 std. dev.) represent the average emission factor of the replicates. The calculated variability in terms of relative standard deviations of the replicates was ±11–22% (BC), ±10–31% (NOx) and ±5–22% (HC) of the mean values. The variability may have been slightly larger for the EGR sweeps due to the shorter averaging time. The variability of repeated EGR scans of hydrotreated vegetable oil (HVO) can be found in Appendix A (Figure A3).

### 3.1.2. Organic Aerosol and PAH Emissions

The particulate phase organic aerosol (OA), including the PAH concentrations, was estimated from the AMS and compared to eBC deduced from the aethalometer. The ratios of OA to eBC mass (OA/eBC) as a function of intake $O_2$ concentration are presented in Figure 2a. The OA/eBC ratios display a minimum at approximately 12–13% intake $O_2$ concentration. The highest OA concentrations with respect to eBC were found at very high EGR (intake $O_2$ concentration <10%), similarly to the

gas-phase HC emissions (Figure 1b). The contribution from lubricating oil to the organic aerosol is high at a high $O_2$ concentration (low EGR) (see, for example, [47] for emissions from the same engine). The emitted OA is originating from different processes at low and high levels of EGR. At low $O_2$ concentrations (high EGR), a less efficient combustion and a slower soot formation process results in the formation and emission of combustion-derived emission compounds that partition to the particle phase (for example PAHs). At high $O_2$ concentrations (low EGR), a high degree of complete combustion, high temperatures, and high availability of $O_2$ result in strongly suppressed eBC emissions and low amounts of combustion-derived compounds being emitted. At high $O_2$ concentrations, the OA/eBC therefore increases as a result of less eBC emissions and a relatively constant generation of PM originating from the lubricating oil. A larger fraction of the OA found in particles at high $O_2$ concentrations is therefore linked to lubrication oil, and not directly to the combustion process or fuel.

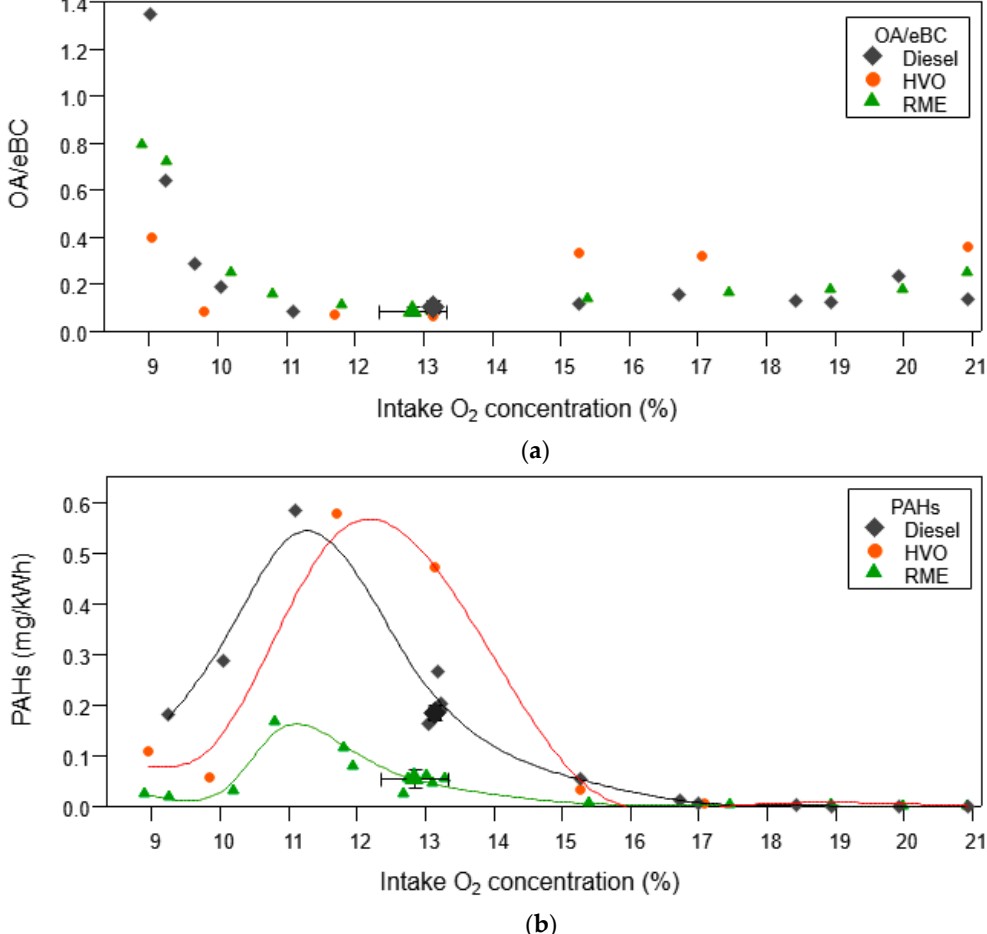

**Figure 2.** (**a**) The organic aerosol to equivalent black carbon ratio (OA/eBC) and (**b**) polycyclic aromatic hydrocarbons (PAHs) emission factors (mg/kWh) vs. exhaust gas recirculation (EGR) expressed as intake $O_2$ concentration. The lines shown in (**b**) are interpolated values and are only included to guide the eye. PAH and OA concentrations were assessed from the aerosol mass spectrometer (AMS) data. The OA fraction increase for all fuels as the $O_2$ concentration decrease at high levels of EGR (i.e., $O_2 < 11\%$). The RME fuel strongly decreases the PAH emissions compared to HVO and diesel, independently of EGR. The data in Figure 2 are from single EGR sweeps with relatively short averaging time in each point (~1.5 min), except at 13% intake $O_2$ where repeated measurements for diesel and RME was performed (displayed with error bars of ±1 std. dev.). The calculated variability in terms of relative standard deviations of the replicates was ±25% (OA/eBC) and ± 10–36% (PAHs) of the mean values. The variability may have been slightly larger for the EGR sweeps due to the shorter averaging time.

The PAH emission factors (mg/kWh) were assessed from the AMS measurements and show a maximum at around 11–12% $O_2$ (Figure 2b) for all fuels. For RME, the PAH emissions were strongly reduced compared to diesel for all levels of EGR. For HVO there was an increase at 13% intake $O_2$, but at other $O_2$ concentrations there was no difference or reduction in PAH emission by HVO compared to diesel. Repeated measurements at 13% intake $O_2$ during filter collection showed that the PAH emissions (AMS) were increased by 78% (95% CI (222%, 57%)) for HVO in comparison to diesel, while RME reduced PAH emissions by 83% (95% CI (20%, 145%)) compared to diesel. These results are qualitatively in line with the emission factors from the offline GC-MS PAH analysis performed on the PM extracts which also showed an increase with HVO (although only 35%) and a strong decrease with RME (Section 3.2). It should be noted that at the highest and lowest studied levels of EGR, the particle sizes were reduced (Figure A4). This may lead to a reduced collection efficiency of the AMS, thus PAH emissions in these points may have been underestimated. A detailed PAH analysis using GC-MS is presented and discussed in Section 3.2.

HVO is a fuel that has essentially zero aromatic and zero oxygen content. HVO resulted in reduced eBC emissions but slightly increased PAH emissions compared to diesel at 13% intake $O_2$. Reduced BC emissions have been found in several previous studies [56–58]. Other studies have found a slight increase in total PAH concentration for blends with HVO compared to other types of biodiesel [59] even though the HVO fuel is free of aromatics. We hypothesize that this can be attributed to its homogeneous composition and few large compounds associated with high vaporization enthalpies. This could result in combustion at more fuel lean conditions (higher air/fuel ratio), with reduced eBC but unaltered PAH emissions relative to combustion with petroleum diesel.

RME is a fuel that has essentially zero aromatic content, but ~10% oxygen content. RME resulted in further reduced eBC emissions and strongly reduced PAH emissions compared to diesel. The most likely explanation is that when oxygen from the fuel is added to fuel-rich regions during combustion, this leads to the further reduction of eBC and a strong reduction in PAH emissions. The reason why RME shows a stronger reduction in PAH emissions than in eBC emissions requires further study.

### 3.1.3. Thermal Optical Analysis (OC/EC) and Relationship to PM Measurements

Figure 3 presents the results of the thermal-optical carbon analysis conducted for the diesel (10%, 12%, 13% and 17% intake $O_2$), HVO (13% intake $O_2$) and RME (13% intake $O_2$) in comparison to eBC emissions (MSS) normalized to total carbon (TC) content. The TC content is the sum of elemental carbon (EC), refractory organic carbon (rOC) and non-refractory organic carbon (nrOC). Particles were sampled from the dilution tunnel on quartz filters (no methanol extraction).

For 12%, 13% and 17% intake $O_2$ concentrations, the organic carbon (OC) fraction was dominated by nrOC. At 10% intake $O_2$, a substantial amount (~1/2) of the OC was characterized as rOC, here defined as the sum of OC4 and pyrolytic carbon (in EUSAAR 2 protocol). Smaller amounts of rOC were also observed for the high temperature diesel sample at 17% intake $O_2$ concentration (low EGR).

The eBC/TC ratio is also presented in Figure 3. The eBC concentrations measured by the MSS are overall in relatively good agreement with the elemental carbon (EC) concentrations. However, at low and high levels of EGR, the eBC measurement is a poor measure of the total carbon mass. The results suggest that neither the nrOC nor rOC were detected with the MSS (photoacoustic detection). The use of eBC to infer particle mass may to some extent be challenged by these results since the rOC compounds are likely to survive heating/evaporation in standardized test methods for solid particle diesel emissions.

The high fraction of rOC at 10% intake $O_2$ (simulating LTC) is in agreement with studies performed with model flames that show a transition from EC to rOC for lowered combustion temperatures [60]. This suggests that the change in soot properties and PAHs observed in model flames and burners are also relevant to LTC in engines. Whether the oxidation reactivity, as well as the surface structure of soot with a high rOC fraction, differs from eBC or EC needs to be studied further. This may affect the particle removal efficiency in a diesel particulate filter (DPF).

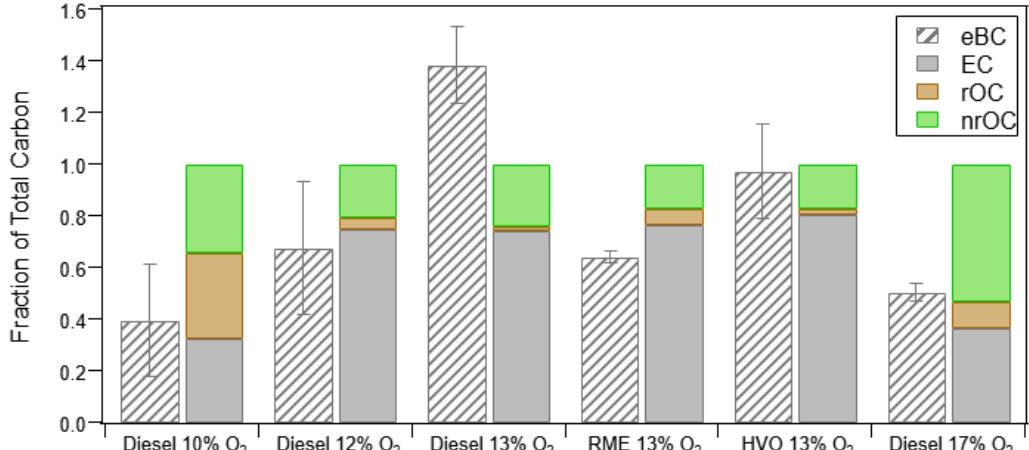

**Figure 3.** equivalent Black Carbon (eBC), elemental carbon (EC), refractory organic carbon (rOC), non-refractory organic carbon (nrOC) mass emissions normalized to the total carbon concentration (the sum of EC, rOC and nrOC) for the different intake $O_2$ concentrations and fuels. Black carbon (eBC) concentrations with ± 1 std. dev. were simultaneously measured online with the Photo-acoustic technique (MSS). The carbon content of the filter collected aerosol mass was measured with the thermal-optical analysis, and classified into three different categories, elemental carbon (EC) and organic carbon (OC) subdivided into two categories. The OC categories were non-refractory OC (nrOC) that evolves at temperatures up to 450 °C, and refractory OC (rOC) that evolves at temperatures exceeding 450 °C (in inert He) or detected in an oxidizing atmosphere (pyrolytic carbon). The nrOC corresponds to the sum of OC1, OC2 and OC3, rOC correspond to the sum of OC4 and pyrOC in the EUSAAR_2 protocol. The total carbon content is the sum of EC, rOC and nrOC. Non-refractory OC (especially in the OC1-OC2 channels) may have been overestimated due to positive artefacts from adsorption of gas phase OC on the quartz filters (no back-up filter was used). This would have affected samples with low collected particle masses the most, in this case diesel 13% and diesel 17%.

### 3.1.4. Particle Size Distributions and TEM Analysis

EGR was the main variable influencing the particle mobility size distributions (Figure A4 in Appendix A). Two modes could be distinguished in the particle size distributions, an accumulation mode and in a few cases a nucleation mode. Particle mobility size distributions sampled simultaneously with the filter replicates can be found in Appendix A (Figure A5a,b). The GMD of the accumulation mode was reduced at both low and high EGR compared to medium levels of EGR. The nucleation mode dominated in number concentration at operation without or with very low EGR, and at very high levels of EGR ($O_2$ < 9%) for all fuels. For the particle collection of diesel at 17% intake $O_2$ concentration, the nucleation mode was 30–50% of the total number concentration with a GMD of 26 nm. The nucleation mode number concentration for the other particle collections (diesel at 10% intake $O_2$, and diesel, HVO and RME at 13% intake $O_2$) was <5% of the total number concentration.

The average number concentration per gram of collected PM and the accumulation mode GMD are given in Table 2. The average number concentration per collected mass was an order of magnitude higher for diesel at 10% and 17% intake $O_2$ compared to diesel at 13% intake $O_2$. RME and HVO had a factor ~2 higher number concentration per collected mass compared to diesel at the 13% intake $O_2$. The fuel effect on accumulation mode size distributions was significant; at 13% intake $O_2$ a GMD of 70 ± 3 nm was found for RME compared to 90 ± 5 nm for HVO and 104 ± 8 nm for diesel. The smaller GMD for RME may be a kinetically driven reduction in agglomeration rates due to reduced eBC concentrations, but could also be related to the altered soot formation or soot oxidation processes.

The particle mobility size distributions (Figure A5b) suggest that the eBC and particle mass (Table 3) reduction for HVO was linked to a decrease in both mobility size and the particle number (PN) emissions. For RME, the data suggest that the particle mass was reduced primarily as a result of smaller particles, rather than a significantly reduced PN.

**Table 2.** The geometric mean of the soot primary particle size measured from the TEM images, the estimated specific surface area (SSA), the mobility size of the aggregates (geometric mean diameter ±1 std. dev. in the time series) measured with the DMS and the average number concentration per gravimetric PM mass. The GSD for the primary particle size and SSA distributions were 1.32–1.50. For primary particle size and SSA, intervals in brackets represent the 95% confidence interval of the distribution parameters.

| | Diesel | | | | HVO | RME | Printex90 |
|---|---|---|---|---|---|---|---|
| Average Intake $O_2$ Concentration (%) | 10 | 12 | 13 | 17 | 13 | 13 | - |
| Primary particle size (GMD; nm) | 22.0 (20.7, 23.4) | 18.2 (16.3, 20.3) | 17.4 (16.2, 18.8) | 16.1 (14.9, 17.5) | 20.9 (19.1, 22.8) | 15.0 (13.7, 16.4) | 14.5 (13.7, 15.4) |
| Estimated SSA ($m^2$/g) | 152 (143, 161) | 183 (164, 204) | 191 (177, 206) | 207 (191, 224) | 160 (146, 174) | 222 (203, 243) | 230 (217, 243) |
| Aggregate size (GMD; nm) from DMS | 55 ± 9 | 90 ± 5 | 104 ±8 | 62 ± 4 | 90 ± 5 | 70 ± 3 | - |
| Average number emissions (#/g PM) | $1.3 \times 10^{16}$ | - | $3.6 \times 10^{15}$ | $1.1 \times 10^{16}$ | $6.3 \times 10^{15}$ | $8.8 \times 10^{15}$ | - |

**Table 3.** Chemical composition of the extracted PM samples. The polycyclic aromatic hydrocarbons (PAHs) are given in groups of native PAHs and PAH derivatives, a full list of all measured compounds can be found in Table A4.

| | Diesel | | | HVO | RME |
|---|---|---|---|---|---|
| Average Intake $O_2$ Concentration (%) | 10 | 13 | 17 | 13 | 13 |
| PM (gravimetric; mg/kWh) | 64 ± 11 | 386 ± 61 | 8 ± 2 | 133 ± 15 | 144 ± 27 |
| OC/TC | 0.65 | 0.12 | 0.40 | 0.28 | 0.32 |
| PAHs * (µg/g) | | | | | |
| Native PAHs | 23,700 | 2470 | 858 | 9960 | 1180 |
| Alkyl-PAHs | 400 | 483 | 77 | 644 | 150 |
| DBTs | 47 | 78 | 128 | 86 | 94 |
| Oxy-PAHs | 2490 | 1450 | 314 | 2630 | 596 |
| Nitro-PAHs | 131 | 21 | 8 | 65 | 40 |
| Total all PAHs | 26,800 | 4500 | 1390 | 13,400 | 2060 |
| Total all PAHs (µg/kWh) | 1514 | 954 | 7 | 1330 | 170 |
| Sum BaPeq (µg/g) | 4685 | 165 | 59 | 1067 | 60 |
| Metals (2 most abundant) | | | | | |
| Fe (µg/g) | 220 | 137 | 2120 | 247 | 116 |
| Cu (µg/g) | 2350 | 629 | 13,200 | 1630 | 2290 |
| Total all metals (µg/g) | 2820 | 905 | 15,500 | 1990 | 2530 |
| Total all metals (µg/kWh) | 180 | 350 | 121 | 266 | 364 |

* The extended measurement uncertainty of the analysis is <15%.

Figure 4 presents TEM images for samples from diesel at 10%, 12%, and 17% intake $O_2$ concentration (Figure 4a–c), and for diesel, HVO and RME at 13% intake $O_2$ concentration (Figure 4d–f). The average primary particle sizes range from 15 nm to 22 nm, with the smallest size found for the particles from combustion of RME (Table 2). Estimations of the specific surface area (SSA) are presented in Table 2. The SSA calculation assumes that each primary particle in the agglomerates only has point contact; hence, the whole surface area of each primary particle contributes to the total surface area. The sample collected at 10% intake $O_2$ concentration has a more fused structure and point contact between the primary particles is a less valid assumption. It is therefore likely that the derived SSA overestimates the true surface area. The more fused structure indicates that the soot agglomeration occurs at different stages in the in-cylinder soot evolution process between the combustion at low and higher $O_2$ concentrations. On the other hand, the inherent material density has been shown to increase with soot maturity level [61]. The choice of a constant material density for all samples may therefore bias the SSA, making it too low for the less mature soot collected at 10% intake $O_2$.

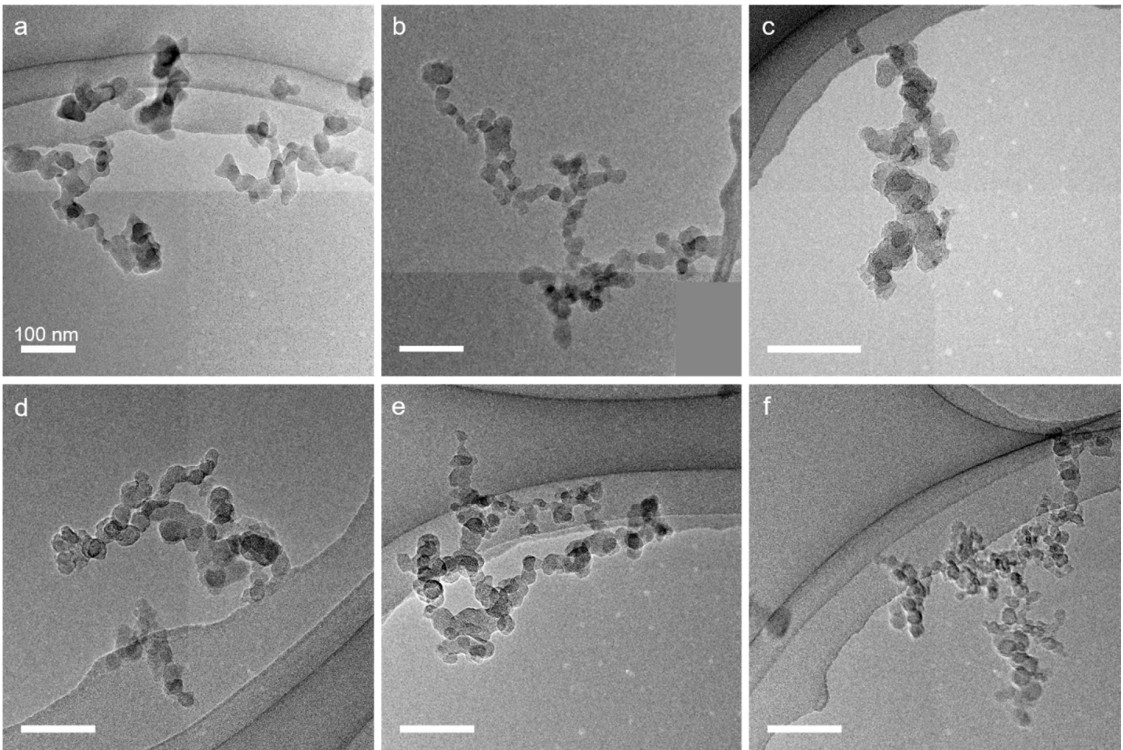

**Figure 4.** Representative TEM images of samples from diesel (**a**–**d**), HVO (**e**) and RME (**f**). The particles are sampled at different levels of exhaust gas recirculation (EGR), i.e., intake $O_2$%; (**a**) 10, (**b**) 12, (**c**) 17, (**d**) 13, (**e**) 13 (HVO) (**f**) 13 (RME).

### 3.2. Analysis of Extracted PM for PAHs and Metals

The composition of the extracted particles used for the ROS analysis is presented in Table 3. A full list of all measured native PAHs and PAH derivatives are given in Table A4. The total particulate PAH mass fraction was highest for the diesel sample collected at 10% intake $O_2$ (26,800 µg/g), followed by the HVO sample at 13% intake $O_2$ (13,400 µg/g). The lowest PAH fractions were found for RME at 13% intake $O_2$ (2060 µg/g) and diesel at 17% intake $O_2$ (1390 µg/g). The offline analysis showed slightly higher PAH emissions for HVO (1330 µg/kWh) than diesel (954 µg/kWh) when quantifying the total PAH emissions in µg/kWh. In agreement with the online AMS analysis, the RME showed strongly reduced PAH emissions compared to both diesel and HVO combustion. Additionally, the sample collected at 10% intake $O_2$ had a higher fraction of larger native PAHs and the highest mass fraction of nitro-PAHs (131 µg/g) by far. The Benzo(a)Pyrene equivalent (BaPeq) toxicity was estimated using toxicity equivalent factors for the PAHs. The BaPeq concentration was, by far, the highest for the 10% intake $O_2$ sample with 4685 µg/g, compared to 59–1067 µg/g for the other samples, indicating a higher genotoxic potential.

The mass fraction of oxy-PAHs (Quinones) was highest for the 10% intake $O_2$ sample (2490 µg/g) and HVO at 13% intake $O_2$ (2630 µg/g). Oxy-PAHs have been associated with strong acellular ROS production [62]. DBTs showed a completely different dependence: no clear effect of fuel, but a clear increase in concentration (µg/g) with increasing temperature (increasing intake $O_2$ concentration). The obtained difference in composition may affect the ROS formation potential and toxicity of the particles. The diesel samples showed similar to the AMS analysis (Figure 2b), showing that PAH emissions were strongly reduced with increasing intake $O_2$ concentration (low EGR). This can potentially be explained by a more rapid conversion of PAHs to BC as combustion temperature increases with a decreasing EGR level [63] for the petroleum diesel.

The most abundant metals found were Cu and Fe, and the highest metal mass fractions were found in the diesel 17% intake $O_2$ sample, 2120 and 13,200 µg/g respectively (Table 3). Metal analysis

of new and used lubricating oil used in the same setup showed that Cu and Fe increased in the used oil while other metal additives remained constant [64]. This indicates that the Cu and Fe content originated from in-cylinder wear, accumulated in the lubricating oil, and exited the engine with the exhaust particles.

When the total metal emission factors (μg/kWh) were calculated, only moderate variations were found between the five combustion conditions. The variations were within a factor of three for diesel as function of intake $O_2$ and within a factor of 1.5 for the three fuels at constant intake $O_2$ concentration. This suggests that metal emissions were associated with the lubricating oil rather than with the fuel or the combustion conditions, and shows that the accumulation of Cu and Fe in the lubricating oil with time did not introduce measurable bias in the evaluated metal content of the PM. The different formation pathways for metal-containing compounds and soot suggest that the metal mass fraction in the emissions depends primarily on the total PM emission levels. The renewable diesels had 2.7 to 3.0 times lower PM emissions compared to diesel, according to the gravimetric analysis. The metal fractions in these emissions were measured to be 2.0–2.6 times higher compared to diesel. At 17% intake $O_2$ the PM emissions were around 40 times lower than at 13% intake $O_2$ for diesel. The measured metal mass fraction was, in this case, almost 20 times higher, thus the 17% intake $O_2$ sample had by far the highest mass-fraction of metals (1.5%) in the emitted particles. The toxicity potential induced by the PM metal content may thus be heavily weighted towards combustion conditions with low eBC emissions, but remain similar when compared to the engine energy output (i.e., metal emissions μg/kWh).

## 3.3. ROS Formation Potential of Diesel Exhaust Particles

All samples produced ROS in comparison to control treatments (0 μg/mL). None of the samples showed autofluorescence at a level that would interfere with the assay. One sample (diesel 10% intake $O_2$) showed about 1.2% of the ROS signal coming from autofluorescence; the other five samples showed <0.1% autofluorescence. As reference, Printex90 (P90), which is a potent ROS forming carbon black that has been linked to in vivo DNA-damage in animal models [51] was used.

The 17% intake $O_2$ diesel sample generated the highest ROS formation with PM mass being the dose metric and the 10% intake $O_2$ sample generated by far the lowest ROS formation (Figure 5). The samples generated at almost identical combustion conditions (13% intake $O_2$) had similar ROS formation potential for diesel and RME. HVO showed a slight elevation in ROS production (Figure 5), although this difference disappeared at higher doses (not shown). The slope of the linear curves ($\alpha$; fluorescence per μg PM) in Figure 5 was used as the measure of particle ROS formation potential. All samples showed lower ROS formation compared to P90, which is in qualitative agreement with previous comparisons of standard diesel exhaust particle NIST SRM 1650 and P90 [65].

Studies have shown that the Brunauer–Emmett–Teller (BET) surface area is an important dose metric for pulmonary toxicity induced by spherical nanoparticles, including carbonaceous material, in animal models [66] and in vitro studies [67]. Stoeger et al. (2009) showed a linear relationship between the BET surface area and acellular oxidative potential for manufactured carbon black particles, including P90, flame soot and a standard reference diesel exhaust sample (NIST SRM 1650). We tested the hypothesis that the specific surface area (SSA) of the soot agglomerates would be linearly related to the ROS formation potential. The ROS formation potential was therefore compared to the SSA estimated from the TEM images of the soot primary particles (Figure 6). The line connecting P90 with the origin represents the hypothetical relationship between soot SSA and ROS formation for the P90. The PM samples all showed less ROS formation per SSA than P90. The ROS formation potential of the diesel sample at 10% intake $O_2$ was 10% of the ROS formation potential for P90 when adjusted for SSA, and 36–69% of P90 for the other samples (Figure 6). This shows that all samples except the diesel 10% intake $O_2$ sample are potent ROS inducers.

The higher ROS formation for P90 compared to diesel soot can stem from a number of particle-induced mechanisms. The exhaust particles are not only pure solid carbon particles, but have additional components of organics and metals. Stoeger et al. (2009) found a lower oxidative potential

for diesel exhaust particles compared to bare carbon black particles and flame soot with low organic fraction [16]. The organics condensed on the particles may reduce the ROS formation in two ways: (1) condensation of organics leads to a loss of physical surface area as the pores in the aggregate are filled; (2) condensed organics may shield the active solid surface so it is not accessible for ROS formation [16,68].

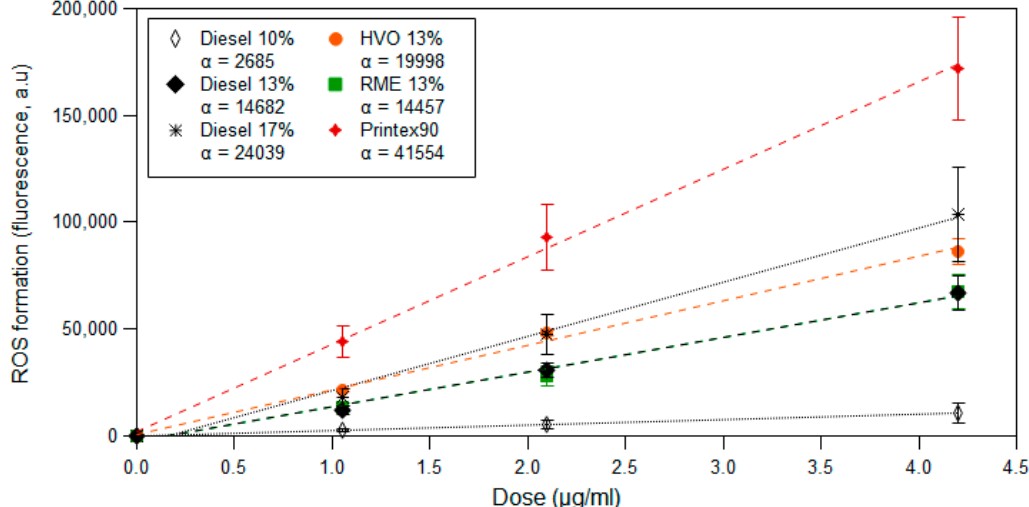

**Figure 5.** The dose response reactive oxygen species (ROS) generation of the samples with error bars of ±1 standard error of the mean. Only the lowest doses are displayed to show the linear dose-response relationship. The slope of these lines is used as a measure of the ROS formation potential per mass unit ($\alpha$) of each material. The ROS curves reach a plateau (not shown) at higher doses due to depletion of reactant ($2',7'$-dichlorodihydrofluorescein diacetate ($DCFH_2$-DA) probe), hence the slope of the linear regression is based on the first data points where linearity is observed.

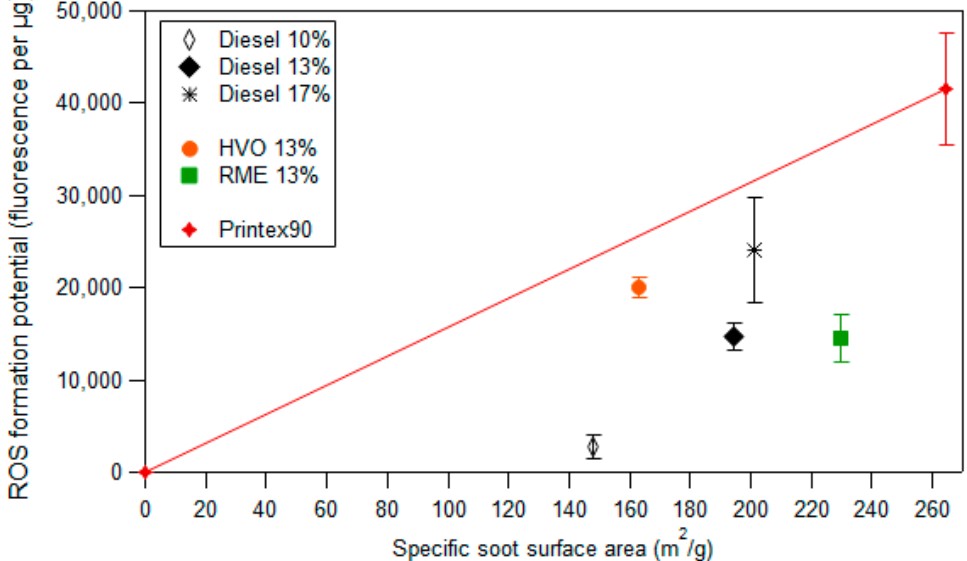

**Figure 6.** The reactive oxygen species (ROS) formation potential per mass unit (with error bars of ±1 standard error of the mean) versus the soot specific surface area ($m^2$/g). The red line shows the hypothetical ROS formation efficiency for materials with the same ROS formation efficiency per specific surface area (SSA) as Printex90. The ROS formation potential of the particle samples were compared to the Printex90 SSA adjusted potential. Diesel 10% (intake $O_2$) showed only 10% of this potential, while diesel and RME at 13% intake $O_2$ had 43% and 46% respectively. The HVO (13% intake $O_2$) and diesel at 17% intake $O_2$ had the highest ROS formation potential with 69% and 64% respectively.

Other studies have linked ROS formation to organic components [69] and metal components. We therefor investigated the correlation between ROS and the chemical composition. Neither the OC, PAH or metal mass fraction correlated well with the ROS formation potential (Figure A6a–c in Appendix A). This suggests that neither of these parameters were the main drivers of ROS. However, metals are known sources of ROS formation, and the influence of additional metals on soot particles would increase rather than decrease ROS formation. Transition metals, especially Cu and Fe, are well-known in metal catalysed Haber–Weiss and Fenton reactions [70] and transition metals can catalyse ROS reactions both on the particle surface and as soluble ions [71]. Therefore, it is important to consider if these metals could be responsible for some of the observed ROS formation for the diesel 17% intake $O_2$ sample, since this case has the highest metal content by far. Tests with CuO (15 nm, PlasmaChem, Berlin, Germany) in a similar ROS assay setup as used here showed that CuO produced 20% more ROS than P90 (EU project H2020 No 760840, GRACIOUS) (Unpublished data, contact: Nicklas R. Jacobsen (nrj@nfa.dk)). Since the summed mass fraction of metals of the diesel sample collected at 17% intake $O_2$ is only about 1.5%, the suggested influence on the total ROS production is very low and not the cause for the high ROS production of this sample. In agreement with this, P90 induces ROS in absence of metal contamination [53]. Biswas et al. (2009) showed that removing organic components from engine exhaust particles using a thermal denuder reduced the oxidative potential (OP) of the particles by 50–100% using the DTT assay, thus a significant fraction of the OP could be attributed to the organic fraction [72]. This is different from our findings. Perhaps the discrepancy may be associated with different ROS-probes used in the two studies, with the DTT assay being more sensitive to ROS formed from organic PM constituents.

*3.4. Framework to Interpret Particle ROS Formation Potential Based on Fundamental Soot Formation and Oxidation Mechanisms in Combustion*

In this section, we discuss how the soot properties, including surface properties, change during combustion by first considering the soot formation process and then the soot oxidation (removal) process in the engine.

Increased EGR and reduced $O_2$ availability lead to significantly reduced ROS formation potential per mass (Figure 7a). The linear relation and strong correlation ($R^2 = 0.87$) between ROS formation potential and intake $O_2$ concentration indicates that the in-cylinder $O_2$ availability and flame temperatures are fundamental combustion parameters that affect the ROS generating ability of diesel soot particles. The ROS formation potential of HVO (13% intake $O_2$) was higher than for RME and diesel. The organic aerosol emissions of HVO (at 13% intake $O_2$) was, to a higher degree, characterized by low molecular weight oxy-PAHs (p-Fluorenone; 9,10-Anthraquinone). These can contribute to ROS formation when measured with the DCFH assay. On the other hand, similar mass fractions of oxy-PAHs were present in the diesel sample collected at 10% intake $O_2$, which showed a low ROS formation potential.

The immature soot formed at low temperature conditions (10% intake $O_2$) was distinct from the other samples and contained large fractions of refractory OC and non-refractory PAHs. Noticeably, this soot had by far the lowest ROS production potential, both per SSA and per mass unit. Immature soot has distinctly different surface properties compared to mature soot. Stoeger et al. (2009) denoted immature soot produced at low temperature conditions in a mini-CAST flame soot generator as "High OC" and found that it had a substantially lower oxidative potential (based on the consumption of ascorbate), than predicted by the BET surface area. Our results suggest that low oxygen, low temperature conditions in modern diesel engine concepts could similarly to the low temperature mini-CAST soot have lower ROS forming potential than soot formed at high combustion temperatures. We recommend further studies to test possible relationships between combustion temperatures and the ROS forming ability and toxicity of soot.

In the fuel-lean post-flame region, soot oxidation is dominated by the readily available $O_2$ [73]. When the main heat release of combustion has occurred in the diesel combustion process, the soot

formation is negligible but the soot oxidation continues. During this stage, soot primary particles, as well as aggregated structures, shrink or become removed by complete oxidation. It is well-known that the surface functional groups change in the soot oxidation phase [61]. The type of oxygenated surface functional groups that are present at soot surfaces depend on the oxidation process. In the late diesel combustion and exhaust, phenols (C–OH) are more abundant than carbonyl (C=O) groups because of the higher thermodynamic stability to continued (complete) oxidation of soot to $CO_2$ [74,75]. The late cycle diesel combustion is an efficient oxidation reactor which may influence soot surface properties [47,54,76].

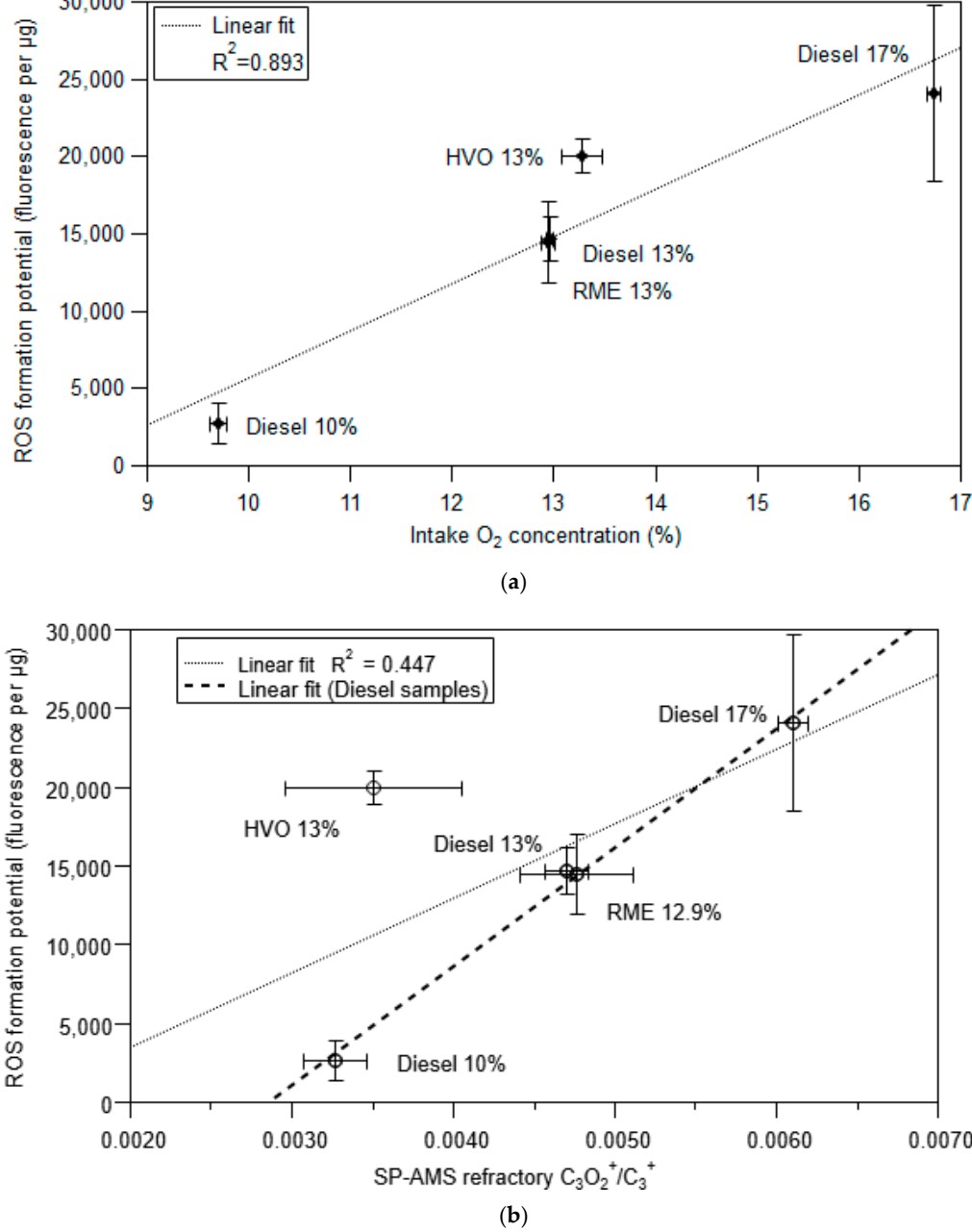

**Figure 7.** (**a**) Relationship between the reactive oxygen species ROS formation potential per mass unit (with error bars of ±1 standard error of the mean) and intake $O_2$ concentration (i.e., EGR), and (**b**) relation between the ROS formation potential and the soot particle aerosol mass spectrometer (SP-AMS) refractory $C_3O_2^+/C_3^+$ fragment ratio. a) Error bars of ±1 std. dev. of the intake $O_2$ concentration (**b**) Error bars for $C_3O_2^+/C_3^+$ show ±1 standard error.

Both intake $O_2$ concentration and the generated ROS were found to be correlated to the SP-AMS signal intensity ratio of refractory $C_3O_2^+$ to $C_3^+$ fragments (Figure 7b). $C_3^+$ is a marker for mature rBC. $C_3O_2^+$ fragments in the SP-AMS appear strictly for laser heating/vaporization. We here hypothesize that $C_3O_2^+$ fragments represent oxygenated surface functional groups which are strongly chemically bonded to soot surfaces and situated at edge-sites on the graphitic lamellae of carbon. This has previously been suggested for $CO^+$ and $CO_2^+$ that are oxygen-containing fragments commonly observed in the SP-AMS mass spectra of soot [77]. However, the $CO^+$ and $CO_2^+$ fragments can also have a non-refractory origin from OA which induce uncertainty in the quantification of their refractory signals.

These results presented in Figure 7a,b indicate that the ROS forming ability of diesel soot can increase with higher formation temperatures both as a result of more mature (rBC-like) soot, and due to increased partial oxidation on the soot particle surfaces. Additional studies with a more sensitive chemical characterization of the surface oxygen functionalization are required to assess the influence from in-cylinder oxidation on the soot particle ROS forming ability.

## 4. Conclusions

The physicochemical properties and emission levels without any aftertreatment devices were investigated as a function of EGR (intake $O_2$ level) and renewable fuel substitution in a highly controlled heavy-duty diesel engine. The study design was developed to test the influence of intake $O_2$ concentration (EGR) and combustion temperatures on particle emission characteristics and toxicity.

Both the emission levels and the physicochemical particle properties were found to depend strongly on the $O_2$ availability (the amount of EGR) and thus flame temperatures in the combustion cylinder. HVO and RME resulted in a general reduction of eBC emissions by 30% and 54% respectively in comparison to petroleum (MK1) diesel at 13% intake $O_2$. The reduction of eBC and PM may be primarily from the reduced particle mobility size at a similar number of emissions for the RME case, while the mass reduction was due to both lower PN emissions (#/kWh) and reduced mobility size for HVO. RME generated the lowest PAH emissions in both, in terms of mass fraction and in absolute emission levels. High levels of EGR (diesel 10% intake $O_2$) resulted in immature soot emissions characterized by refractory organic carbon (rOC) and non-refractory PAHs. Fe and Cu detected in exhaust PM were found to originate from engine wear and were enriched in the lubricating oil. Precautions should be taken to not use alloys or additives in the engine or lubricating oil which potentially could be toxic if they are aerosolized and inhaled.

The soot samples did not show a correlation between the ROS formation potential and metal or PAH content or SSA. Moreover, the ROS formation potential was not significantly increased or reduced by neither renewable HVO nor RME when compared to diesel at the same engine operating condition. However, as the PM mass emission levels were two to three times lower for the renewable fuels compared to diesel, the ROS formation related to fuel consumption (or engine power output) was lower for the renewable fuels compared to diesel.

The ROS formation potential was correlated to the intake $O_2$ concentration (i.e., EGR) and to refractory oxygen-containing fragments in the SP-AMS. This indicates that engine operating conditions, combustion temperatures and the availability of $O_2$ are important engine parameters that can alter the soot emission ROS formation potential, and ultimately the related diesel engine PM toxicity potential. In addition, the SP-AMS results point towards a correlation between the amount of strongly bound surface oxygen functional groups and the soot ROS formation potential. Therefore, based on our results, we propose that the main mechanisms controlling the soot ROS formation potential is (1) a soot maturation process from rOC to rBC that occurs with increasing combustion temperature, and (2) due to late-cycle soot oxidation and the addition of strongly bonded oxygenated functional groups. These observations suggest that further research on the ROS formation potential and carcinogenesis of solid, carbonaceous PM emitted from combustion engines should be directed towards detailed analysis of soot maturity (e.g., the carbon nanostructure) and surface properties. The possibility that surface oxygen

functionalization may control part of the ROS formation potential may suggest that aftertreatment, such as diesel oxidation catalysts, could potentially modify the ROS formation potential.

The study was limited to a single engine operation at low load (IMEPg 6 bar) and constant speed (1200 rpm). Therefore, it is not clear to what extent these results can be extrapolated to other engine operating conditions and real-world driving, for example, high load and transient conditions as well as emissions that have passed exhaust aftertreatment systems. The assessment of the ROS formation potential was carried out on particles that were collected on Teflon filters followed by extraction in methanol. To minimize possible alterations in particle properties from the filter collection and extraction procedure, future studies may apply alternative methods that allow particle collection into liquids for direct ROS formation studies. Additionally, the response of different diesel soot components in various types of ROS probes is of interest for future evaluations.

**Author Contributions:** Overall idea and design of experiments: J.P., U.V., L.G., M.T., V.B.M. Design of aerosol characterization experiments and data interpretation: L.G., J.P., V.B.M., A.C.E. Aerosol data collection and analysis: L.G., V.B.M. TEM analysis and particle extraction: L.G. Selection/design of engine operation and set points: M.T., P.C.S. Engine operation and collection of engine data: P.C.S., S.S. Analysis of engine data: S.S. Design, data collection and analysis of ROS measurements N.R.J. Interpretation of ROS data based on physicochemical properties: L.G., V.B.M., N.R.J., J.P., K.M.B., U.V. Development of method for PAH derivatives: B.S., A.M.K. Collection of PAH data and analysis: Y.J.E., A.M.K. Design of metal analysis, data collection and data analysis: K.L. L.G. drafted the original manuscript with major input from V.B.M. and J.P. All co-authors contributed to writing and/or critical revisions. All authors have read and agreed to the published version of the manuscript.

**Funding:** This research was financed by the Swedish Research Councils FORMAS (2016-00697), Vetenskapsrådet (2018-04200) and AFA Insurance (160323).

**Acknowledgments:** Flemming Cassee, Dan Leseman, Aneta Wierzbicka and Christina Isaxon are acknowledged for contributing to selecting the particle extraction and weighing methods and valuable discussions. Karin Lovén is acknowledged for the OC/EC analysis. Katrin Loeschner thanks Agilent for providing the Agilent 8900 ICP-QQQ instrument.

**Conflicts of Interest:** The authors declare no conflict of interest.

## Appendix A

*Appendix A.1. Method*

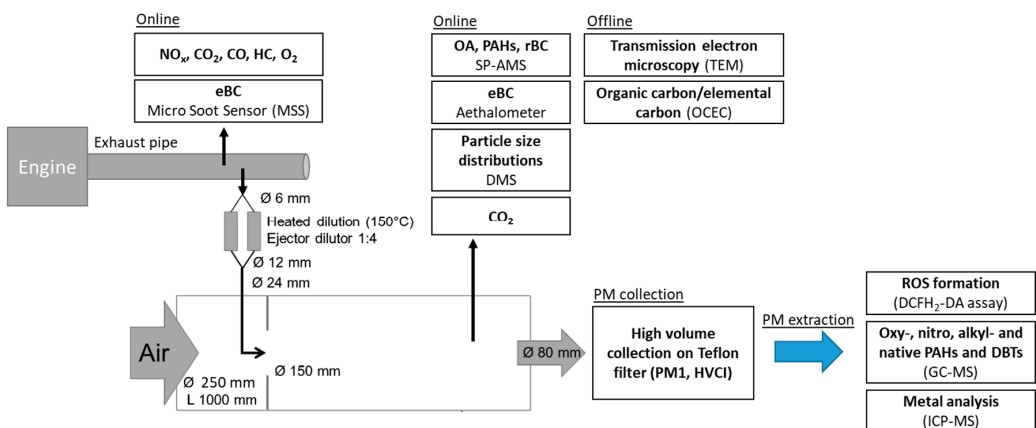

**Figure A1.** The dilution system (partial dilution flow tunnel) for the PM collection.

Appendix A.1.1. SP-AMS

The SP-AMS is an online instrument and was operated with a time-resolution of 10 s. The SP-AMS was used to investigate and quantify the chemical composition of the non-refractory organic and refractory carbonaceous PM. To investigate the non-refractory composition, the SP-AMS was run in the standard aerosol mass spectrometer (AMS, Aerodyne Inc., Billerica, USA) mode with thermal desorption on a heated tungsten target (600 °C) and subsequent electron ionization (70 eV) and detection

in a high-resolution time-of-flight mass spectrometer. The detection of highly refractory components is possible through the incorporation of an ND:YAG (1064 nm) laser which heats infrared-absorbing particles (such as black carbon) to boiling temperatures, where rapid vaporization occurs. The vapors are subsequently ionized and detected identically to the standard AMS. The SP-AMS was set up to record mass spectra between approximately 10–900 Da. PAHs have low fragmentation probabilities and were treated specifically. To assess the PAH content of the organic aerosol, the contribution from seven common PAHs in sizes between 202–300 Da (or m/z) were assessed and multiplied by a factor of four. These PAHs are commonly the strongest contributors to the total PAH signal for soot aerosols, and have parent peak signals at 17–35% of the total PAH concentration [78]. It is not possible to determine the specific isomer related to a given mass to charge in the aerosol mass spectra (e.g., m/z 202). For this, offline analysis was used when filter sampling and extraction was conducted. In addition, the signal intensities of refractory SP-AMS carbon fragments, $C_3^+$, $CO_2^+$, and $C_3O_2^+$ that originate from the soot particles were analysed.

Appendix A.1.2. PAH Analysis

Chemicals and Reagents

All adsorbents, silica gel 60 (Merck, Darmstadt, Germany) and sodium sulfate (Merck, Darmstadt, Germany) were cleaned by thermal treatment at 450 °C and activated at 100 °C before use. All solvents were of glass distilled quality (Merck, Darmstadt, Germany). Two deuterated internal standard mixtures (1 ng μL$^{-1}$) containing the 16 U.S. Environmental Protection Agency (US-EPA) priority PAHs (Dr. Ehrenstorfer (Augsburg, Germany) and a mixture of four nitro-PAHs (Cambridge Isotope Laboratories, Inc., Andover, MA, USA) were used. Four native mixtures, all at 1 ng L$^{-1}$, containing 16 US EPA PAHs (Dr. Ehrenstorfer, Augsburg, Germany), 16 alkylated species (Ultra Scientific, North Kingstown, RI, USA), 17 nitro-PAHs and nine oxy-PAHs (Dr. Ehrenstorfer, Augsburg, Germany), and six dibenzothiophenes (DBTs) (Toronto Research Chemicals (TRC), Toronto, Ontario, Canada) were used for the detection and quantification of target compounds. Octachlorornaphthalene (OCN) (Ultra Scientific, North Kingstown, RI, USA) (1 ng L$^{-1}$) was used as recovery standard (RS). Quality controls (QCs) (diesel particles NIST SRM 2975) were purchased from the US National Institute of Standards and Technology (NIST) (Gaithersburg, MD, USA).

Sample Extraction, Clean-Up and Analysis

The extracted PM were spiked in glass vials with a portion of 40 μL of the two internal standard mixtures, respectively, and then extracted by sonication for 3 h in 3 mL of dichloromethane at maximal amplitude in a Sonica Ultrasonic Extractor (Soltec, Milan, Italy). Following extraction, the samples were cleaned up using a Pasteur pipet, with a small plug of wool in the bottom, filled with 2 cm silica powder and some sodium sulfate on top. The elute was evaporated under nitrogen flow until only a third of the initial volumes was left, solvent exchanged using n-hexane (ca 3 mL), and finally evaporated to ca 200 μL. The samples were transferred to GC glass inserts vials (Agilent Technologies) and 40 μL of RS was added and samples were reduced to a small volume (ca 30–40 μL) for analysis.

In total, 32 PAHs—17 nitro-PAHs, nine oxy-PAHs and six DBTs—were analysed. Tables A1–A3 shows the transitions, limits of detection (LODs), as well as retention times (RT) for all investigated compounds. Target compounds were separated on an Agilent 5975C mass spectrometer (MS) coupled to a 7890A gas chromatograph (GC, Agilent Technologies). The samples (2 μL) were injected using an Agilent autosampler unit. The capillary column used was a DB-5MS (30 m × 0.25 mm, 0.25 μm, Agilent Technologies). Helium was the carrier gas at a flow rate of 1.0 mL/min. The temperature program was as follows: initial temperature 50 °C for 3 min; ramp at 10 °C/min to 180 °C and held for 5 min; ramp at 3 °C/min to 300 °C and held for 20 min; injection at oven temperature at 250 °C; and transfer line at 250 °C. Electron impact ionization (EI) was performed for PAHs, alkylated PAHs and DBTs, at 70 eV energy and at a 230 °C ion source temperature. Electron capture negative chemical ionization

mode (ECNCI) was used for the nitro- and oxy-PAH species. The quadrupole temperature was 150 °C. The MS was operated in selected ion monitoring mode (SIM) for both EI and ECNCI modes.

**Table A1.** Transitions and retention times of PAHs, alkylated PAHs, deuterium labeled internal standards (IS) and recovery standard (RS) used. Analysis was performed, using electron impact ionization.

| Name | m/z | RT | IS | m/z | RT |
|---|---|---|---|---|---|
| naphthalene | 128 | 11.95 | naphthalene-$d_8$ | 136 | 11.90 |
| 2-methylnaphthalene | 142 | 13.61 | acenaphthylene-$d_8$ | 160 | 15.74 |
| 1-methylnaphthalene | 142 | 13.83 | acenaphthylene-$d_8$ | 160 | 15.74 |
| biphenyl | 154 | 14.77 | acenaphthylene-$d_8$ | 160 | 15.74 |
| 2,3-dimethylnaphthalene | 156 | 15.59 | acenaphthylene-$d_8$ | 160 | 15.74 |
| acenaphthylene | 152 | 15.74 | acenaphthylene-$d_8$ | 160 | 15.74 |
| acenaphthene | 154 | 16.18 | acenaphthene-$d_{10}$ | 164 | 16.10 |
| 2,3,5-trimethylnaphthalene | 170 | 17.27 | acenaphthene-$d_{10}$ | 164 | 16.10 |
| fluorene | 166 | 17.65 | fluorene-$d_{10}$ | 176 | 17.56 |
| 1-methylfluorene | 180 | 19.92 | fluorene-$d_{10}$ | 176 | 17.56 |
| phenanthrene | 178 | 21.88 | phenanthrene-$d_{10}$ | 188 | 21.75 |
| anthracene | 178 | 22.17 | anthracene-$d_{10}$ | 188 | 22.08 |
| 2-methylphenanthrene | 192 | 24.99 | phenanthrene-$d_{10}$ | 188 | 21.75 |
| 1-methylphenantrene | 192 | 25.74 | phenanthrene-$d_{10}$ | 188 | 21.75 |
| 1-methylanthracene | 192 | 25.82 | phenanthrene-$d_{10}$ | 188 | 21.75 |
| 3-methylphenanthrene | 192 | 25.83 | phenanthrene-$d_{10}$ | 188 | 21.75 |
| 2-phenylnaphthalene | 204 | 27.20 | phenanthrene-$d_{10}$ | 188 | 21.75 |
| fluoranthene | 202 | 29.73 | fluoranthene-$d_{10}$ | 212 | 29.61 |
| pyrene | 202 | 31.24 | pyrene-$d_{10}$ | 212 | 31.12 |
| 1-methylfluoranthene | 216 | 33.83 | pyrene-$d_{10}$ | 212 | 31.12 |
| retene | 234 | 34.06 | pyrene-$d_{10}$ | 212 | 31.12 |
| 1-methylpyrene | 216 | 35.5 | pyrene-$d_{10}$ | 212 | 31.12 |
| benzo(a)anthracene | 228 | 40.44 | benzo(a)anthracene-$d_{12}$ | 240 | 40.29 |
| chrysene | 228 | 40.68 | chrysene-$d_{12}$ | 240 | 40.49 |
| 2-methylchrysene | 228 | 40.68 | chrysene-$d_{12}$ | 240 | 40.49 |
| benzo(b)fluoranthene | 264 | 47.96 | benzo(b)fluoranthene-$d_{12}$ | 242 | 43.89 |
| benzo(k)fluoranthene | 264 | 48.19 | benzo(k)fluoranthene-$d_{12}$ | 252 | 48.12 |
| benzo(a)pyrene | 252 | 50.14 | benzo(a)pyrene -$d_{12}$ | 264 | 49.99 |
| perylene | 252 | 50.69 | benzo(a)pyrene-$d_{12}$ | 264 | 264 |
| indeno(1,2,3-c,d)pyrene | 276 | 56.86 | indeno(1,2,3-c,d)pyrene-$d_{12}$ | 288 | 57.03 |
| dibenzo(a,h)anthracene | 276 | 57.20 | dibenzo(a,h)anthracene-$d_{14}$ | 288 | 57.03 |
| benzo(g,h,i)perylene | 276 | 58.19 | benzo(g,h,i)perylene-$d_{12}$ | 288 | 58.05 |
| | | | octachloronaphthalene (OCN, RS) | 404 | 49.38 |

**Table A2.** Transitions and retention times of dibenzothiophenes (DBTs) and deuterium labeled internal standards (IS) and recovery standard (RS) used. Analysis was performed, using electron impact ionization (EI).

| Name | m/z | RT | IS | m/z | RT |
|---|---|---|---|---|---|
| dibenzothiophene | 184 | 21.50 | phenanthrene-$d_{10}$ | 188 | 21.75 |
| 2-methyldibenzothiophene | 198 | 24.21 | phenanthrene-$d_{10}$ | 188 | 21.75 |
| 1-methyldibenzothiophene | 198 | 24.76 | phenanthrene-$d_{10}$ | 188 | 21.75 |
| 4-methyldibenzothiophene | 198 | 25.38 | phenanthrene-$d_{10}$ | 188 | 21.75 |
| 2,8-dimethyldibenzothiophene | 212 | 28.20 | fluoranthene-$d_{10}$ | 212 | 29.61 |
| 2,4,7-trimethyldibenzothiophene | 226 | 30.95 | fluoranthene-$d_{10}$ | 212 | 29.61 |

**Table A3.** Transitions and retention times of nitrated (nitro-PAH) and oxygenated (oxy-PAH) PAHs and deuterium labeled internal standards (IS) and recovery standard (RS) used. Analysis was performed using capture negative chemical ionisation mode (ECNCI).

| Name | m/z | RT | IS | m/z | RT |
|---|---|---|---|---|---|
| nitro-PAHs | | | | | |
| 1-nitronaphthalene | 173 | 17.82 | 2-nitrofluorene-$d_9$ | 220 | 30.80 |
| 2-nitronaphthalene | 173 | 18.59 | 2-nitrofluorene-$d_9$ | 220 | 30.80 |
| 5-nitroacenaphthalene | 199 | 27.87 | 2-nitrofluorene-$d_9$ | 220 | 30.80 |
| 2-nitrofluorene | 211 | 30.81 | 2-nitrofluorene-$d_9$ | 220 | 30.80 |
| 9-nitroanthracene | 223 | 31.49 | 2-nitrofluorene-$d_9$ | 220 | 30.80 |
| 9-nitrophenanthrene | 223 | 33.65 | 2-nitrofluorene-$d_9$ | 220 | 30.80 |
| 3-nitrofluoranthene | 247 | 42.62 | 3-nitrofluoranthene-$d_9$ | 256 | 42.70 |
| 4-nitropyrene | 247 | 42.98 | 6-nitrochrysene-$d_9$ | 284 | 50.70 |
| 1-nitropyrene | 247 | 43.99 | 6-nitrochrysene-$d_9$ | 284 | 50.70 |
| 2-nitropyrene | 247 | 44.54 | 6-nitrochrysene-$d_9$ | 284 | 50.70 |
| 7-nitrobenz(a)anthracene | 273 | 48.67 | 6-nitrochrysene-$d_9$ | 284 | 50.70 |
| 6-nitrochrysene | 273 | 50.72 | 6-nitrochrysene-$d_9$ | 284 | 50.70 |
| 3-nitronbenzanthrone | 275 | 50.88 | 6-nitrochrysene-$d_9$ | 284 | 50.70 |
| 1,3-dinitropyrene | 292 | 52.42 | 6-nitrochrysene-$d_9$ | 284 | 50.70 |
| 1,6-dinitropyrene | 292 | 53.71 | 6-nitrochrysene-$d_9$ | 284 | 50.70 |
| 1,8-dinitropyrene | 292 | 54.59 | 6-nitrochrysene-$d_9$ | 284 | 50.70 |
| 6-Nitrobenzo(a)pyrene | 297 | 57.73 | 6-nitrochrysene-$d_9$ | 284 | 50.70 |
| oxy-PAHs | | | | | |
| naphthalene-1-aldehyde | 156 | 16.38 | 2-nitrofluorene-$d_9$ | 220 | 30.80 |
| 2-naphthaldehyde | 156 | 16.42 | 2-nitrofluorene-$d_9$ | 220 | 30.80 |
| p-fluorenone | 180 | 20.57 | 2-nitrofluorene-$d_9$ | 220 | 30.80 |
| phenanthrene-9-aldehyde | 206 | 31.23 | 2-nitrofluorene-$d_9$ | 220 | 30.80 |
| 9,10-anthraquinone | 208 | 26.92 | 2-nitrofluorene-$d_9$ | 220 | 30.80 |
| 1,4-anthraquinone | 208 | 29.01 | 2-nitrofluorene-$d_9$ | 220 | 30.80 |
| benzo(a)fluorene | 230 | 37.57 | 3-nitrofluoranthene-$d_9$ | 256 | 42.70 |
| benzo(b)fluorene | 230 | 39.21 | 3-nitrofluoranthene-$d_9$ | 256 | 42.70 |
| benzanthrone | 230 | 41.34 | 3-nitrofluoranthene-$d_9$ | 256 | 42.70 |
| benz(a)anthracene-7,12-dione | 258 | 43.73 | 3-nitrofluoranthene-$d_9$ | 256 | 42.70 |

*Appendix A.2. Results*

Variability

Three EGR sweeps on three different days (i.e., repeated experiments) were conducted for HVO (Figure A3a,b). In order to estimate the variability of eBC due to replicate measurements, a lognormal function was fit to all data points (Figure A3a). Consideration of measurement uncertainties (±1 std. dev.) was included in the fitting procedure. The shaded area represents the 95% confidence interval of the fit, and can be regarded as a constraint on the uncertainty. It is clear that the relative uncertainty is higher for lower and higher $O_2$ compared to medium $O_2$ concentrations.

The variability of $NO_x$ due to replicate measurements was low and is seen in Figure A3b.

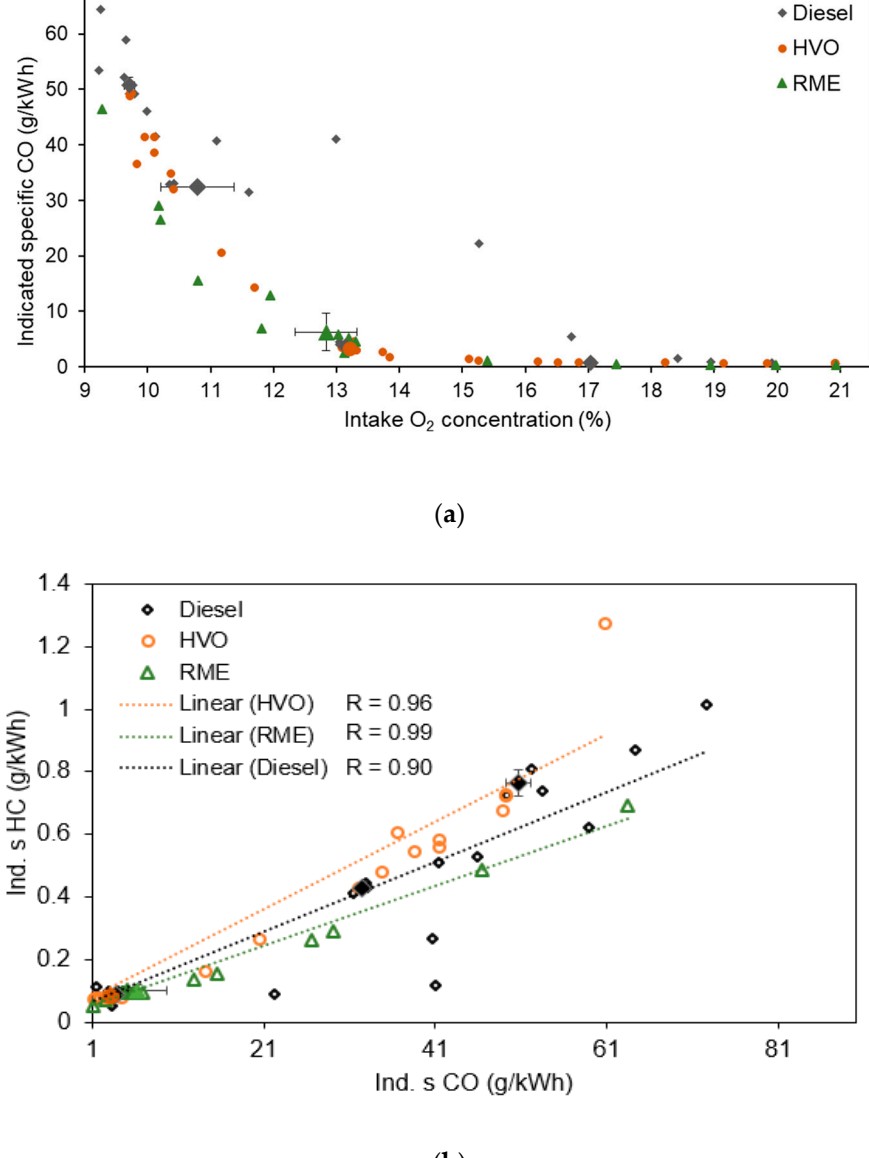

(**a**)

(**b**)

**Figure A2.** (**A**) The CO emissions (g/kWh) for the fuels depending on EGR (expressed as intake $O_2$ concentration). Markers with error bars (±1 std. dev.) represent the average emission factor of the replicates. We approximate the variability of the replicates to ±5–55% of the mean values. (**B**) The HC-CO correlation with error bars (±1 std. dev.) of the replicates.

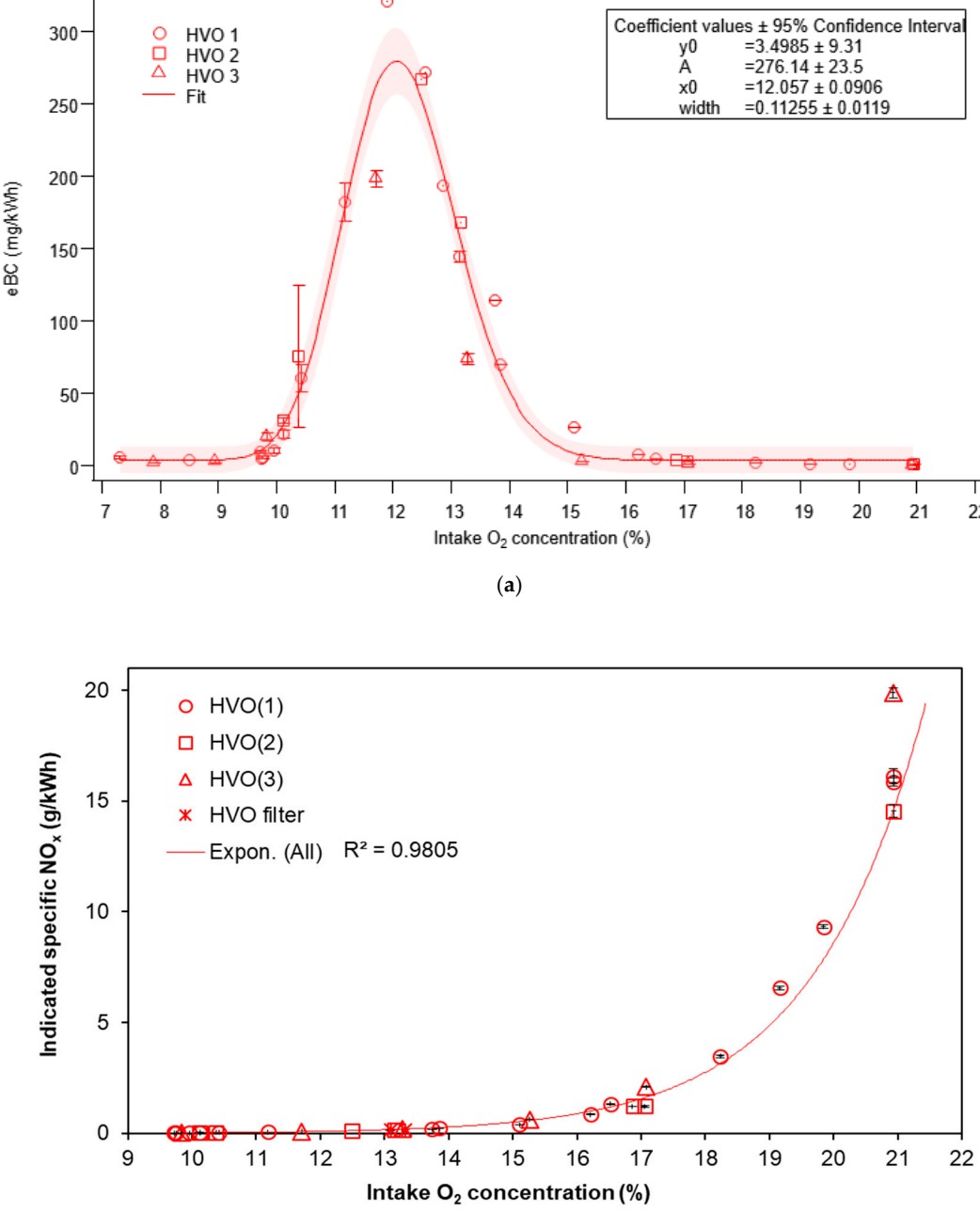

**Figure A3.** The (**a**) eBC emissions (g/kWh) and b) NO$_x$ (g/kWh) of EGR sweeps on three different days (i.e., repeated experiments) of HVO depending on EGR (expressed as intake O$_2$ concentration). Error bars represent the measurement variabilities (±1 std. dev.) for the sampling periods. A (**a**) lognormal function and (**b**) exponential function was fit to all data points. Consideration of measurement uncertainties (±1 std. dev.) was included in the fitting procedure. The shaded area in (**a**) represents the 95% confidence interval of the fit, and can be regarded as a constraint on the uncertainty.

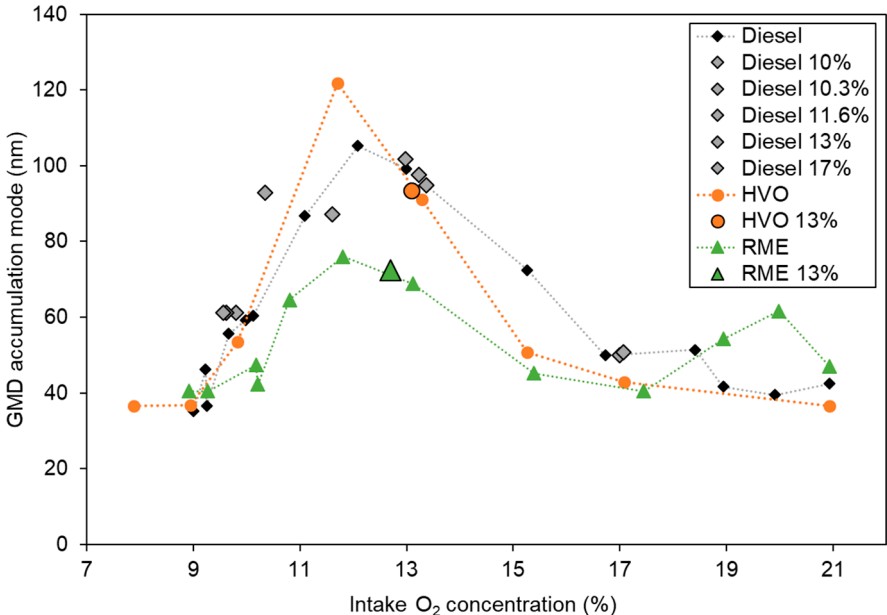

**Figure A4.** The accumulation mode GMD (nm) for the fuels depending on EGR (expressed as intake $O_2$ concentration).

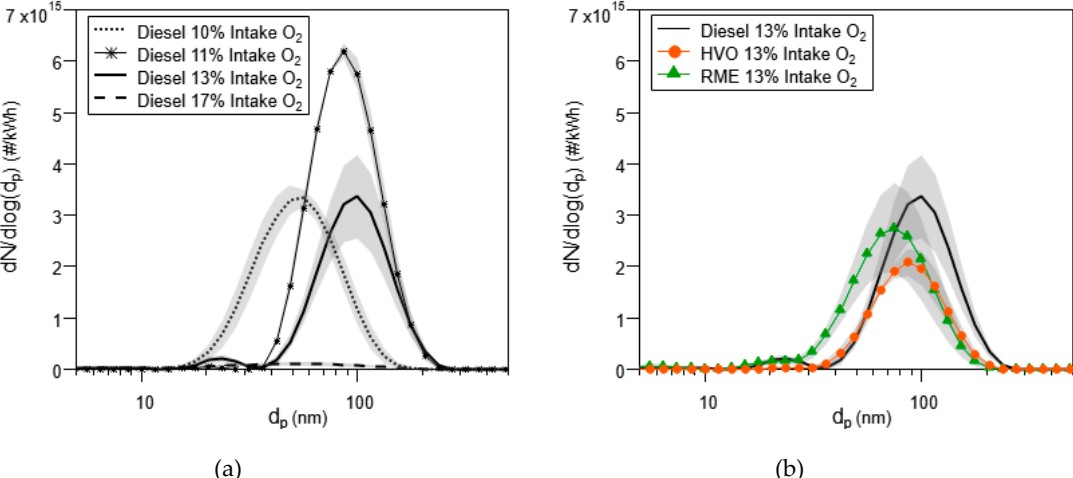

|  |  |
|:---:|:---:|
| (a) | (b) |

**Figure A5.** The average particle mobility size distribution of the filter replicates, the shadowed area represent ±1 std. dev. (**a**) The diesel samples generated at different levels of EGR (expressed as intake $O_2$ concentration) and (**b**) HVO and RME compared to diesel generated at 13% intake $O_2$. The average particle number (PN) emission was similar for RME and diesel at 13% intake $O_2$, while the PN emission was reduced around a factor 1.6 for HVO compared to diesel.

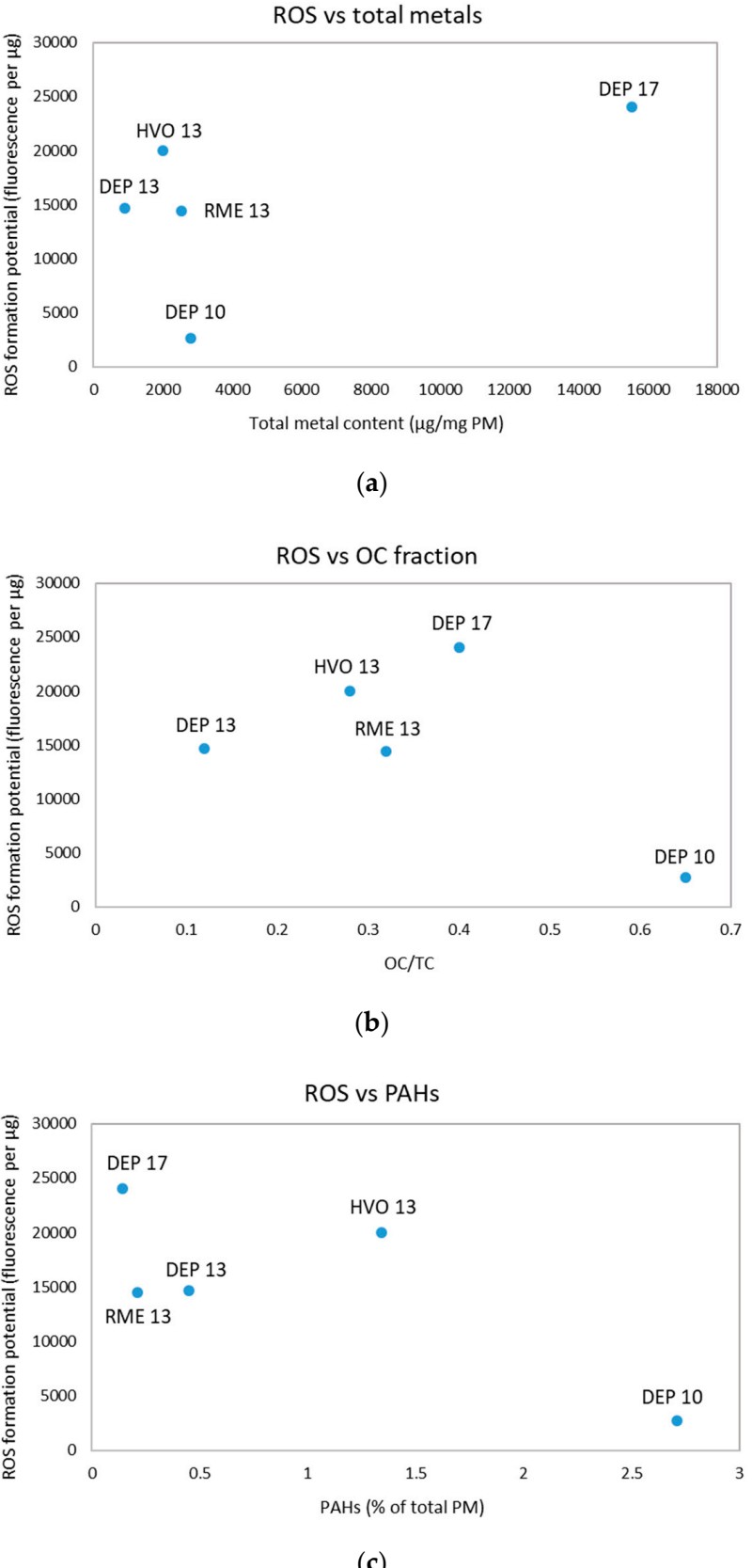

**Figure A6.** The ROS formation potential per mass versus the (**a**) total metal content. (**b**) the OC/TC fraction and (**c**) total PAH concentration. No clear relationship was found for these parameters to ROS formation potential.

**Table A4.** Measured content of native PAHs and PAH derivatives in diesel emission particles and NIST2975 reference material (µg/g particles). PAH levels of NIST2975 quantified after soxhlet extraction or PFE at 100–200 °C, as reported by NIST, are given in brackets for comparison.

| | DEP | | | HVO | RME | Reference |
|---|---|---|---|---|---|---|
| | MK1 Low-Sulfur Diesel | | | Hydrotreated Vegetable Oil | Rapeseed Methyl Ester | NIST2975 |
| | 10 | 13 | 17 | 13 | 13 | |
| Native PAH | | | | | | |
| naphthalene | 4.4 | 136.8 | 6.8 | 52.8 | 19.3 | 1.4 (1.8) |
| biphenyl | 5.0 | 67.7 | 2.7 | 63.2 | 3.5 | 0.3 (0.3) |
| acenaphthylene | 16.2 | 96.9 | 2.8 | 68.7 | 5.7 | <LOD |
| acenaphthene | 1.0 | 17.0 | 2.0 | 3.5 | 0.5 | 0.1 |
| fluorene | 17.5 | 44.8 | 2.9 | 62.5 | 5.1 | 0.5(0.4) |
| phenanthrene | 423.7 | 236.9 | 44.7 | 1167.4 | 207.8 | 10.6(17.3) |
| anthracene | 60.9 | 17.8 | 3.3 | 47.9 | 21.4 | 0.1(0.0) |
| fluoranthene | 1677.0 | 141.4 | 162.2 | 1591.4 | 325.2 | 23.3 (26.9) |
| pyrene | 2381.6 | 182.8 | 223.2 | 2341.4 | 293.1 | 0.8 (0.9) |
| retene | 9.1 | 271.8 | 166.1 | 120.6 | 157.6 | <LOD |
| benzo(a)anthracene | 1378.15 | 5.5 | 15.2 | 187.7 | 6.3 | 0.2 (0.3) |
| chrysene | 2609.2 | 9.3 | 52.8 | 369.9 | 15.8 | 3.9 (4.6) |
| benzo(b)fluoranthene | 4158.16 | 207.6 | 40.7 | 548.4 | 27.8 | 8.2(11.5) |
| benzo(k)fluoranthene | 1094. | 887.6 | 59.0 | 187.9 | 36.2 | 0.3 (0.7) |
| benzo(a)pyrene | 3739.3 | 61.3 | 32.8 | 759.2 | 33.4 | 0.3(0.0) |
| perylene | 638.3 | 14.5 | 10.0 | 137.6 | 2.5 | 0.1(0.0) |
| indeno(1,2,3-c,d)pyrene | 2072.1 | 2.3 | 8.7 | 702.5 | 2.1 | 0.9 (1.4) |
| dibenzo(a,h)anthracene | 40.2 | 27.2 | 7.1 | 49.9 | 4.3 | 0.4(0.3) |
| benzo(g,h,i)perylene | 2608.5 | 31.6 | 14.3 | 1216.6 | 7.6 | 0.9(0.5) |
| coronene | 752.3 | 7.6 | 0.5 | 280.5 | 0.8 | 0.3 |
| SUM Native PAH | 23,686.3 | 2468.7 | 857.6 | 9959.3 | 1176.1 | 52.4 |
| Alkyl-PAH | | | | | | |
| 2-methylnaphthalene | <LOD | 150.3 | 2.8 | 58.3 | 10.4 | 0.8(1.9) |
| 1-methylnaphthalene | 1.4 | 106.5 | 2.5 | 30.8 | 6.4 | 0.4(1.0) |
| 2,3-dimethylnaphthalene | 3.8 | 51.8 | 7.3 | 14.6 | 13.5 | 2.4 |
| 2,3,5-trimethylnaphthalene | 0.2 | 14.7 | 2.1 | 3.3 | 0.8 | 0.2 |
| 1-methylfluorene | 0.4 | 11.1 | 3.3 | 7.8 | 3.3 | 0.1 |
| 4-methylphenanthrene | 8.5 | 16.9 | 7.1 | 15.5 | 9.7 | 0.6 |
| 3-methylphenanthrene | 11.0 | 14.1 | 7.8 | 18.6 | 13.3 | 1.2 (1.0) |
| 1-methylphenanthrene | 11.5 | 16.6 | <LOD | 18.1 | 56.0 | <LOD |
| 1-methylanthracene | 17.6 | 16.6 | 6.9 | 23.3 | 11.9 | 0.6 |
| 2-phenylnaphthalene | 83.1 | 25.5 | 6.3 | 81.1 | 26.9 | 1.1 |
| 1-methylfluoranthene | 41.8 | 23.7 | 6.9 | 98.6 | 17.2 | 0.2 |
| 1-methylpyrene | 166.8 | 34.7 | 23.3 | 268.3 | 30.4 | 0.0 |
| 2-methylchrysene | 54.4 | 0.4 | 1.2 | 6.1 | 0.7 | 0.0 |
| SUM Alkyl-PAH | 400.4 | 482.9 | 77.2 | 644.3 | 150.5 | 7.5 |
| DBT (Dibenzothiophenes) | | | | | | |
| dibenzothiophene | 44.1 | 52.3 | 99.7 | 61.1 | 72.0 | 10.3 |
| 2-methyldibenzothiophene | 1.1 | 4.8 | 13.1 | 8.0 | 6.4 | 0.1 |
| 1-methyldibenzothiophene | 0.4 | 12.4 | 6.6 | 10.6 | 8.4 | 0.2 |
| 4-methyldibenzothiophene | 0.2 | 1.0 | 1.2 | 0.9 | 0.7 | 0.0 |
| 2,8-dimethyldibenzothiophene | 0.5 | 5.9 | 4.0 | 4.1 | 4.5 | 0.0 |
| 2,4,7-trimethyldibenzothiophene | 0.5 | 1.9 | 3.3 | 1.8 | 2.5 | <LOD |
| SUM DBT | 46.8 | 78.2 | 127.8 | 86.3 | 94.5 | 10.5 |
| Nitro-PAH | | | | | | |
| 1-Nitronaphthalene | 0.1 | 1.6 | 0.1 | 2.2 | 3.0 | 0.0(0.0) |
| 2-Nitronapthalene | 0.3 | 11.9 | 0.8 | 22.7 | 13.0 | 0.1(0.1) |
| 5-nitro acenapthalene | 1.3 | <LOD | 2.0 | 0.5 | <LOD | <LOD |

**Table A4.** M*Cont.*

| | DEP | | | HVO | RME | Reference |
|---|---|---|---|---|---|---|
| | MK1 Low-Sulfur Diesel | | | Hydrotreated Vegetable Oil | Rapeseed Methyl Ester | NIST2975 |
| | 10 | 13 | 17 | 13 | 13 | |
| 2-Nitrofluorene | 0.1 | 0.0 | 0.0 | 0.1 | 0.5 | 0.1 |
| 9-Nitroanthracene | 0.6 | 0.0 | 0.0 | 0.1 | <LOD | 1.6(3.0) |
| 9-Nitrophenanthrene | 0.7 | 0.0 | 0.0 | 0.2 | 0.0 | 0.2 |
| 4-Nitropyrene | 9.4 | 0.0 | 0.3 | 3.5 | 1.6 | 0.1(0.2) |
| 3-Nitrofluoranthene | 9.2 | 0.0 | <LOD | 1.2 | 2.1 | 2.2 |
| 1-Nitropyrene | 2.4 | 0.1 | 4.0 | 3.1 | 6.5 | 25.0 |
| 2-Nitropyrene | 88.6 | 0.2 | 0.2 | 26.2 | 1.8 | 0.0 |
| 7-Nitrobenz(a)anthracene | 7.0 | 6.4 | <LOD | 3.7 | 9.4 | 3.2 |
| 6-Nitrochrysene | 0.2 | <LOD | <LOD | 0.0 | 1.7 | 0.8 |
| 3-Nitrobenzanthrone | 0.3 | 0.1 | 0.2 | 0.7 | 0.3 | 0.2 |
| 6-Nitrobenzo(a)pyrene | 10.4 | 0.8 | 0.8 | 1.3 | 0.5 | 0.2(1.4) |
| SUM Nitro-PAHs | 130.5 | 21.2 | 8.4 | 65.5 | 40.4 | 33.7 |
| | Oxy-PAH | | | | | |
| Napthalene-1-aldehyde | 17.7 | 246.9 | 11.9 | 259.9 | 27.5 | 3.0 |
| 2-Naphthaldehyde | 67.6 | 1005.5 | 104.0 | 947.4 | 248.0 | 2.8 |
| p-Fluorenone | 136.0 | 168.3 | 49.7 | 659.8 | 214.5 | 12.7 |
| 9,10 Anthraquinone | 146.6 | 10.5 | 31.9 | 184.8 | 72.6 | 18.0 |
| 1,4 Anthraquinone | 25.2 | 6.5 | 1.3 | 2.7 | <LOD | <LOD |
| Phenanthrene-9-aldehyde | 50.3 | 0.5 | 2.8 | 21.7 | 0.1 | 2.0 |
| Benzanthrone | 1475.6 | 1.3 | 21.6 | 425.3 | 19.2 | 10.3 |
| Benz(a)anthracene-7,12-dione | 201.3 | 0.1 | 0.8 | 8.0 | 2.7 | 13.2 |
| SUM Oxy-PAH | 2488.3 | 1452.3 | 314.3 | 2627.5 | 596.3 | 265.1 |

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
