# Peer review of "Effect of Renewable Fuels and Intake O2 Concentration on Diesel Engine Emission Characteristics and Reactive Oxygen Species (ROS) Formation"

_atmosphere, doi:10.3390/atmos11060641_

Round 1

Reviewer 1 Report

The authors analyzed the emission characteristics of diesel engines under different fuels and different conditions, discussed the factors affecting the ROS formation potential and proposed several hypotheses. Although the results of this study still have uncertainties and repetitions of known information, they are of application value. Therefore, I recommend this manuscript to be published in Atmosphere after revision.

The following are suggestions and comments on the revision of this manuscript.

  1. For the effects of combustion conditions on the emission characteristics from diesel engine, one critical review should be added in the manuscript, Journal of Environmental Sciences, 2020, Vol. 88, pp. 370-384.
  2. I hope to see more details of renewable fuels in the Introduction.
  3. Lines 71 and 73: What’s the meaning of “HC”? Please give the full name when the abbreviation first appears.
  4. Lines 98-99: Has the test been conducted under other engine loads and speeds? Idle speed and unstable driving speed also affect emission characteristics. In order to improve the applicability, various engine conditions should be considered.
  5. Line 101: The full names of HVO and RME have been given previously.
  6. Line 266: Does the low temperature combustion (LTC) here refer to a state of almost no combustion? As shown in Figure 1, both BC and NOx emissions were zero at LTC.
  7. Line 271: It is suggested to explain the meaning of FAME to improve the reader's understanding of the results.
  8. Line 300: Can the ratio of OA/BC represent a performance or a condition of a diesel engine?
  9. Lines 309 and 311: Does combustion-derived and fuel-derived refer to combustion sources and non-combustion sources, respectively?
  10. Line 324: Does “combustion at more fuel lean conditions” mean a sufficient combustion? Please describe this condition in detail.
  11. Lines 444-445: In Lines 310-313, it was mentioned that the low EGR condition inhibited the formation of BC, while the transformation from PAHs to rBC mentioned here was to promote the formation of BC. Is that a contradiction?
  12. Under the same combustion conditions, the emission characteristics of the three fuels varied greatly. Please explain this difference concretely with the nature of those fuels. There is little discussion of their nature in the current results.

Author Response

The authors of the manuscript thanks the reviewers for well-thought and valuable feedback to the manuscript. A thorough revision has been performed in accordance to your feedback which has improved the quality of the manuscript. Point by point answers and changes performed according to the reviewers’ individual comments are provided in separate documents (please see attachment).

In addition to the comments by the reviewers, minor edits have done to improve readability. The main changes are marked in the manuscript, grey for reviewer 1 and blue for reviewer 2.

Reviewer 2 Report

The authors present an interesting study on the effects of fuel properties and engine operating conditions on emissions characteristics and ROS formation. The following are some general comments followed by line-specific comments.

'Introduction' section can include a more comprehensive literature review with a focus on quantitative discussion (some instances are noted in the following comments). 

Different nomenclature like LTC, low NOX medium EGR is can be confusing. Different EGR levels can be referred to in terms of O2 levels. 

Please discuss the limitations of the study.

Limited literature review

26-28: Reduction compared to what?

32-34: Not clear what ‘first-stage’ and ‘second step’ refer to.

42-49: All of the text is qualitative in nature. I suggest quantifying (by citing appropriate references) some information such as the share of diesel exhaust PM in the total radiating forcing and reduction in PM emissions on switching to renewable diesel fuels.

57: Consider removing the word ‘liquid’ as it does not accurately define the state of PAHs.

73-74: Cite the studies being referred to as ‘few studies’

98: Define ‘IMPEG’

149: What is ‘Ch. 2.4’

153: Cite references for the EUSAAR_2 protocol.

222-224: Provide a justification for choosing select PAHs for analysis. If the list does not cover all PAHs, discuss how not characterizing other PAHs might affect the presented data, especially, the ROS potential data.

Overall comment for Section 2: Add a table listing all experiments performed along with critical information such as EGR levels and the number of replicates. Also, it is good to provide information such as calorific value and air/fuel ratio for stoichiometric combustion.

Fig 1a: The x-axis label is confusing. Please edit it for better clarity. It might be useful to mark O2 level corresponding to a stoichiometric ratio of 1 for each fuel.

Figure 1: Why error bars are absent? How many replicates were performed to obtain each data point?

259: Quantify ‘substantial’ because it is not apparent from the graph for range 16-18%.

275: HC does not seem to be lower for RME compared to the other two fuels at 13% O2. Also, error bars are critical when making such statements.

279-280: If experiments were repeated, please report the reduction in average+-standard deviation format.

Figure 1 caption: Provide a correlation plot for CO and HC in supplementary files and quantify the extent of correlation. As the figure caption mentions, some variability data is available. Please plot error bars for all available data points.

298-300: Repetition, and therefore, redundant.

Figure 2 caption: The statement “The OA fraction increase for all fuels as the O2 concentration decrease (i.e. the increasing amount of EGR).” is not always true. HVO OA fraction at 15, 17 and 21 % O2 are higher than that at 13%. Line 301-302 contradicts the figure caption.

304: Quantify ‘slightly’.

316-318: same comment as for 279-280

Figure 3: Because the sum of EC, rOC and nrOC is one, these three components are normalized by the sum of the same three components. However, BC is confusing as it is not clear what is it normalized with and how? In some cases normalized BC is more than one which does not make sense because it is supposed to fraction of total carbon.

356-366: Speculative statement which should be removed.

384: Remove the word ‘accumulation’ to avoid confusion with accumulation mode which is based on loss mechanisms and rates of particles.

385-386: Recommend presenting size distributions in the supplementary file. Also, a general comment about particle size distribution (PSD). PSD evolves via coagulation which is a function of residence time (time before sampling) and conc. (a strong function of dilution factor). Because the dilution factor was varied, comment on its effects on sampled PSD.

426-466: Mention some, if not all, fraction values in the text to avoid referring to tables for each statement.

520: A better review of the previous literature can be added

Section 3.4: This section contains hypotheses not fully or strongly supported by experimental data or previous studies. I recommend shortening and toning down the section by getting rid of speculative components.

Author Response

(The authors gave the same response as above.)

Reviewer 3 Report

The authors study the physicochemical properties and emission levels of petroleum diesel, hydrotreated vegetable oil and biodiesel without any aftertreatment device as function of EGR (intake O2 level) in a highly controlled heavy-duty diesel engine.

This study is one of the most interesting study I have read recently and is very well executed, performed and explained.

I highly recommend to publish it as is.

Author Response

The authors of the manuscript thanks the reviewer for kind feedback of the manuscript. 

A thorough revision has been performed in accordance to the other reviewers' feedback which we think has improved the quality of the manuscript. 

Round 2

Reviewer 2 Report

Thank you for making suggested changes and providing response to my queries.

Author Response

Thank you, particle size distributions are now also included in the supporting information to the manuscript (appendix A, fig A5)